# Multiphasic value biases in fast-paced decisions

Elaine A Corbett[1,2,3]*, L Alexandra Martinez-Rodriguez[3], Cian Judd[1], Redmond G O'Connell[1,2], Simon P Kelly[1,3]*

[1]Trinity College Institute of Neuroscience, Trinity College Dublin, Dublin, Ireland; [2]School of Psychology, Trinity College Dublin, Dublin, Ireland; [3]School of Electrical and Electronic Engineering and UCD Centre for Biomedical Engineering, University College Dublin, Dublin, Ireland

**Abstract** Perceptual decisions are biased toward higher-value options when overall gains can be improved. When stimuli demand immediate reactions, the neurophysiological decision process dynamically evolves through distinct phases of growing anticipation, detection, and discrimination, but how value biases are exerted through these phases remains unknown. Here, by parsing motor preparation dynamics in human electrophysiology, we uncovered a multiphasic pattern of countervailing biases operating in speeded decisions. Anticipatory preparation of higher-value actions began earlier, conferring a 'starting point' advantage at stimulus onset, but the delayed preparation of lower-value actions was steeper, conferring a value-opposed buildup-rate bias. This, in turn, was countered by a transient deflection toward the higher-value action evoked by stimulus detection. A neurally-constrained process model featuring anticipatory urgency, biased detection, and accumulation of growing stimulus-discriminating evidence, successfully captured both behavior and motor preparation dynamics. Thus, an intricate interplay of distinct biasing mechanisms serves to prioritise time-constrained perceptual decisions.

## Editor's evaluation

This study reports important insights into perceptual decisions made under high temporal pressure, by revealing a complex interplay of different biasing mechanisms unfolding during decision formation. The neurally-constrained approach adopted in the study, using the dynamics of EEG correlates of decision-making, provides convincing evidence in favor of the proposed biasing mechanisms. This generic approach could be applied in future work in other contexts, using other types of neural data, to refine, validate or falsify candidate models of human and animal behavior.

**\*For correspondence:**
corbette@ucd.ie (EAC);
simon.kelly@ucd.ie (SPK)

## Introduction

Perceptual decision making is generally well explained by a process whereby evidence is accumulated over time up to a bound that can trigger an action (***Brown and Heathcote, 2008***; ***Link and Heath, 1975***; ***Ratcliff, 1978***; ***Smith and Ratcliff, 2004***; ***Usher and McClelland, 2001***). In most models based on this principle, a given response time (RT) is made up of two temporal components, where the decision variable is either building at a stationary rate ('drift rate') determined by a stable evidence representation, or is suspended, during 'nondecision' delays associated with sensory encoding and motor execution. This simple scheme, developed primarily through the study of perceptual decisions with low-to-moderate speed pressure, affords two ways to explain how faster and more accurate responses are made to higher value or more probable stimuli: through modulating the starting point or drift rate of the process (***Blangero and Kelly, 2017***; ***Feng et al., 2009***; ***Leite and Ratcliff, 2011***;

*Mulder et al., 2012*; *Ratcliff and McKoon, 2008*; *Simen et al., 2009*; *Summerfield and Koechlin, 2010*; *Urai et al., 2019*; *Voss et al., 2004*; *White and Poldrack, 2014*). Corresponding adjustments have been reported in neurophysiological recordings from motor-related areas of the brain (*de Lange et al., 2013*; *Hanks et al., 2011*; *Rorie et al., 2010*). However, recent work has highlighted additional dynamic elements of the decision process whose contributions to choice performance are likely to be accentuated when stimuli require immediate action.

First, when stimulus onset is predictable, anticipatory activity in motor preparation regions can begin to forge a decision even before the stimulus appears. While standard models do allow for anticipatory processing in the setting of the starting point from which the accumulator evolves after sensory encoding, neurophysiological data have revealed that anticipatory motor preparation is often dynamic, proceeding on a trajectory aimed at eventually crossing an action-triggering threshold by itself even in the absence of sensory input (*Feuerriegel et al., 2021*; *Kelly et al., 2021*; *Stanford et al., 2010*). This represents a pre-stimulus signature of a signal identified in neurophysiology studies known as urgency—defined in accumulator models as an evidence-independent buildup component that continues to operate throughout the decision process, adding to sensory evidence accumulation so that the criterion amount of cumulative evidence to terminate the decision reduces with time (*Churchland et al., 2008*; *Hanks et al., 2014*; *Murphy et al., 2016*; *Shinn et al., 2020*; *Steinemann et al., 2018*; *Thura and Cisek, 2014*).

Second, for many suddenly onsetting stimuli, sensory evidence of their distinguishing features emerges some time after the initial sensory neural response signaling their onset (*Afacan-Seref et al., 2018*; *Smith and Ratcliff, 2009*), meaning that detection precedes discrimination. In the case of the widely studied random dot motion stimulus, recent work shows that behavior is well captured by a model in which accumulation begins at the onset of sensory encoding but where it takes a further 400 ms approximately for the direction information to stabilize (*Smith and Lilburn, 2020*). In fact, serial detection and discrimination phases are reflected in human electrophysiological signatures of differential motor preparation during fast, value-biased decisions about other sensory features. Specifically, these signals show biased stimulus-evoked changes initially in the direction of higher value before being re-routed toward the correct sensory alternative (*Afacan-Seref et al., 2018*; *Noorbaloochi et al., 2015*), in line with previously proposed dual-phase models (*Diederich and Busemeyer, 2006*).

Thus, in time-pressured situations decision formation is not suspended until sensory representations stabilize, but rather proceeds through a concerted sequence of anticipatory, detection, and discriminatory processing phases. Although previous work has established the potential importance of these individual phases (*Afacan-Seref et al., 2018*; *Diederich and Busemeyer, 2006*; *Kelly et al., 2021*; *Noorbaloochi et al., 2015*; *Smith and Lilburn, 2020*; *Stanford et al., 2010*), there exists no detailed computational account of how value-biased decision formation dynamics unfold through all three of them. In this study, we used two complementary human electrophysiological signatures of motor preparation during performance of a sudden-onset random dot motion direction discrimination task under a tight deadline, to forge such an account.

We observed a complex pattern of distinct biases exerted across multiple phases including an initial anticipatory buildup in motor preparation for the high-value alternative, a later but steeper anticipatory buildup for the low-value alternative and then, immediately following stimulus onset, a further transient burst toward the high-value alternative. By incorporating urgency signal model components whose initial amplitude and buildup rate were constrained to match the corresponding measures of anticipatory motor preparation, we were able to adjudicate among several alternative multiphase decision process models. We found that a model that featured (1) an initial, transient detection-triggered deflection toward the higher-value alternative and (2) gradually increasing discriminatory sensory evidence, best accounted for behavior, as well as recapitulating the fast dynamics of stimulus-evoked, differential motor preparation. Together, the findings show that, rather than simply enhancing all parameters of the decision process in favor of high-value alternatives, the neural decision architecture has the flexibility to apply biases in opposing directions to different process components, in a way that affords low-value decision signals the chance to 'catch up' when smaller rewards can be attained.

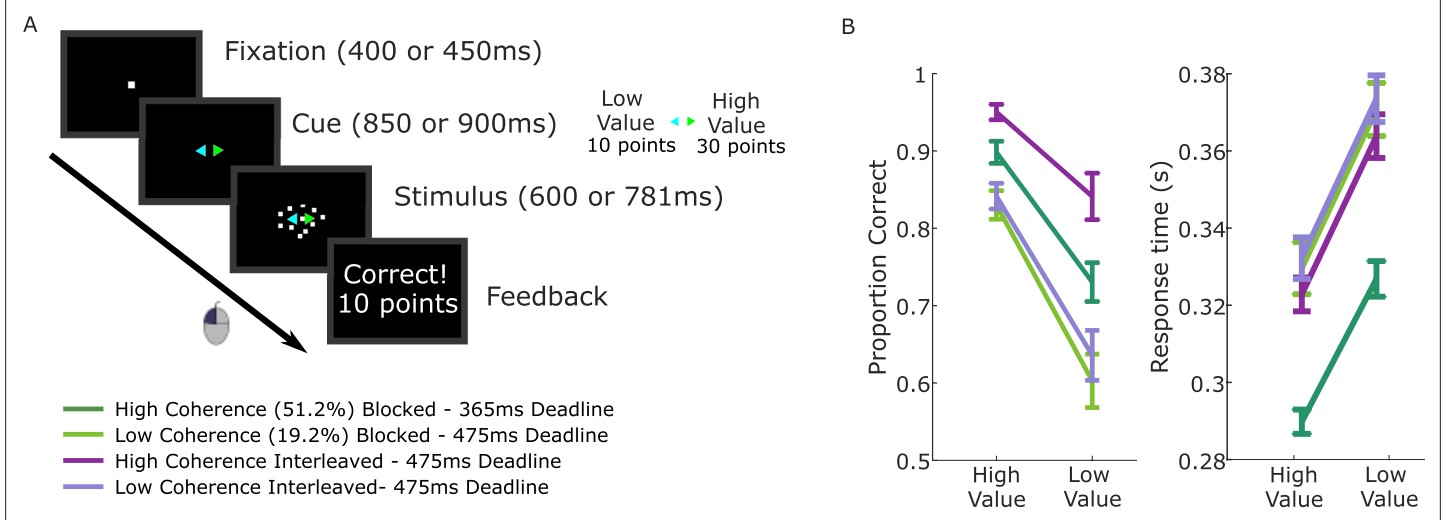

**Figure 1.** Value-cued motion direction discrimination task and behavioral data. (**A**) Trial structure with task conditions below. (**B**) Mean and standard error across participants ($n$ = 17) for proportion correct and median response times (RTs) of correct responses. Repeated measures analyses of variance (ANOVAs) with fixed effects for task condition and value demonstrated that accuracy was higher for high-value trials than low-value trials ($F_{(1,16)}$ = 60.8, $p < 0.001$, partial $\eta^2$ = 0.79), and the median RTs for correct responses were shorter ($F_{(1,16)}$ = 80.7, $p < 0.001$, partial $\eta^2$ = 0.84). In addition to the large value effects, task condition affected accuracy ($F_{(3,48)}$ = 60.2, $p < 0.001$, partial $\eta^2$ = 0.79) and correct RTs ($F_{(3,48)}$ = 38.1, $p < 0.001$, partial $\eta^2$ = 0.61); the high-coherence conditions were more accurate ($p < 0.001$ for blocked and interleaved) and the blocked high-coherence condition, with the shorter deadline, was the fastest ($p < 0.001$ compared to other three conditions). Pairwise comparisons revealed no significant difference between the low-coherence conditions in correct RTs ($p = 0.6$; BF10 = 0.28). The low-coherence interleaved condition was slightly more accurate than the low-coherence blocked condition but not significantly so, and the Bayes factor indicates the data contain insufficient evidence to draw definite conclusions ($p = 0.1$, BF10 = 0.87). The Condition × Value interaction was significant for accuracy ($F_{(3,48)}$ = 6.4, $p = 0.001$, partial $\eta^2$ = 0.29) but not correct RTs ($p = 0.7$).

The online version of this article includes the following figure supplement(s) for figure 1:

**Figure supplement 1.** Quantile–quantile plots.

**Figure supplement 2.** Individual-group quantile correlation ($r^2$) statistics for (**A**) high-value and (**B**) low-value trial distributions.

# Results

## Behavior

Participants performed fast-paced motion direction discrimination using the well-studied random dot kinematogram (RDK) stimulus (***Roitman and Shadlen, 2002***) with a preceding cue indicating the more valuable direction. We recorded scalp electroencephalography (EEG) from seventeen participants performing the task in three blocked regimes: high coherence with a very short deadline; low coherence with a slightly longer deadline; and the two coherences interleaved with the longer deadline (***Figure 1A***). These regimes were similarly challenging but in different ways, allowing us to further explore the extent to which the uncovered value biasing dynamics generalize across task contexts where the demands are placed through lower discriminability versus through tight deadlines, and where stimulus discriminability is heterogeneous versus homogeneous (***Hanks et al., 2011***; ***Moran, 2015***). In each trial, two colored arrows appeared prior to the stimulus onset, the colors of which indicated the respective value of a correct response in each of the two possible directions (left and right). After the onset of the stimulus, participants responded by clicking the mouse button corresponding to the chosen direction with their corresponding thumb. We imposed a single value differential (30 vs. 10 points) that, combined with the deadline and coherence settings, induced a decision-making approach that was guided strongly by both sensory and value information. Correct responses between 100 ms after stimulus onset and the deadline resulted in the points associated with the color cue; otherwise, no points were earned. The value manipulation produced strong behavioral effects across all four conditions, though overall accuracy and RT varied (***Figure 1B***).

Our ultimate goal was to develop a model that could jointly explain the group-average EEG decision signals and behavior. Behavior was quantified in the RT distributions for correct and error responses in each stimulus and value condition, summarized in the 0.1, 0.3, 0.5, 0.7, and 0.9 quantiles

(*Ratcliff and Tuerlinckx, 2002*). Following the analysis of *Smith and Corbett, 2019*, we verified that the individual RT quantiles could be safely averaged across participants without causing distortion by plotting the quantiles of the marginal RT distributions for the individual participants against the group-averaged quantiles, for each of the eight conditions (*Figure 1—figure supplement 1*). The quantiles of the individual distributions were seen to fall on a set of very straight lines, indicating that the quantile-averaged distribution belongs to the same family as the set of its component distributions (*Smith and Corbett, 2019*), thus approximating the conditions for safe quantile-averaging identified by *Thomas and Ross, 1980*. We calculated the Pearson correlations between each individual's quantiles and the group average with that individual excluded, for each condition (see *Figure 1—figure supplement 2*), finding that the lowest $r^2$ was 0.965 while most values were above 0.99. These analyses indicate that quantile-averaging will produce a valid characterization of the pattern of behavioral data in the individuals.

## EEG signatures of motor preparation

Decreases in spectral amplitude in the beta band (integrated over 14–30 Hz) over motor cortex reliably occur with the preparation and execution of movement (*Pfurtscheller, 1981*). When the alternative responses in a decision task correspond to movements of the left and right hands, the signal located contralateral to each hand leading up to the response appears to reflect effector-selective motor preparation that is predictive of choice (*Donner et al., 2009*). Furthermore, before the onset of sensory evidence the 'starting levels' of the signals reflect biased motor preparation when prior expectations are biased (*de Lange et al., 2013*), and are higher under speed pressure for both alternatives (*Kelly et al., 2021*; *Murphy et al., 2016*; *Steinemann et al., 2018*), implementing the well-established decision variable (DV) adjustments assumed in models (*Bogacz et al., 2010*; *Hanks et al., 2014*; *Mulder et al., 2012*). The signal contralateral to the chosen hand then reaches a highly similar level at response irrespective of stimulus conditions or RT, consistent with a fixed, action-triggering threshold (*Devine et al., 2019*; *Feuerriegel et al., 2021*; *Kelly et al., 2021*; *O'Connell et al., 2012*; *Steinemann et al., 2018*). The level of beta before stimulus onset also predicts RT, and its post-stimulus buildup-rate scales with evidence strength, underlining that this signal reflects both evidence-independent and evidence-dependent contributions to the decision process (*Steinemann et al., 2018*). Thus, we can interpret the left- and right-hemisphere beta as reflecting two race-to-threshold motor preparation signals whose buildup trace the evolution of the decision process from stimulus anticipation through to the response (*Devine et al., 2019*; *Kelly et al., 2021*; *O'Connell et al., 2012*).

Here, prior to stimulus onset, motor preparation (decrease in beta amplitude) began to build in response to the value cue, first for the high-value alternative and later for the low-value alternative ($F(1,16)$ = 15.8, p = 0.001, partial $\eta^2$ = 0.5 for jackknifed onsets, *Figure 2A*), and continued to build for both alternatives after stimulus onset. Consistent with prior work suggesting an action-triggering threshold, the signal contralateral to the chosen hand reached a highly similar level at response irrespective of cue-type, coherence, or regime (*Figure 2B*). Before the stimulus onset, rather than generating a stable starting level bias the motor preparation signals continued to increase dynamically. This replicates similar anticipatory buildup observed in a previous experiment with prior probability cues, and does not reflect an automatic priming due to the cue because its dynamics vary strategically with task demands such as speed pressure (*Kelly et al., 2021*). Thus, we take the anticipatory buildup to reflect dynamic urgency that, independent of but in addition to the evidence, drives the signals toward the threshold (*Churchland et al., 2008*; *Hanks et al., 2014*; *Murphy et al., 2016*; *Steinemann et al., 2018*; *Thura and Cisek, 2014*).

We examined this anticipatory activity for evidence of value bias in the period immediately before stimulus onset (*Figure 2A*). Preparation for the high-value alternative was greater than that for the low-value alternative 750 ms after the cue ($F(1,16)$ = 17.6, p < 0.001, partial $\eta^2$ = 0.52). However, despite their later onset, the buildup rates of motor preparation signals for the low-value alternative were significantly steeper (slope from 700 to 800 ms, $F(1,16)$ = 14.7, p = 0.001, partial $\eta^2$ = 0.48), indicating a negative buildup-rate bias. These beta slopes for the high- and low-value alternatives, averaged across conditions, are shown for each individual in *Figure 2—figure supplement 1*. Despite absolute levels of beta amplitude varying quite widely across the group, as is typical in human EEG, the majority of individuals (14 out of 17) show steeper buildup for the low-value alternative. As a

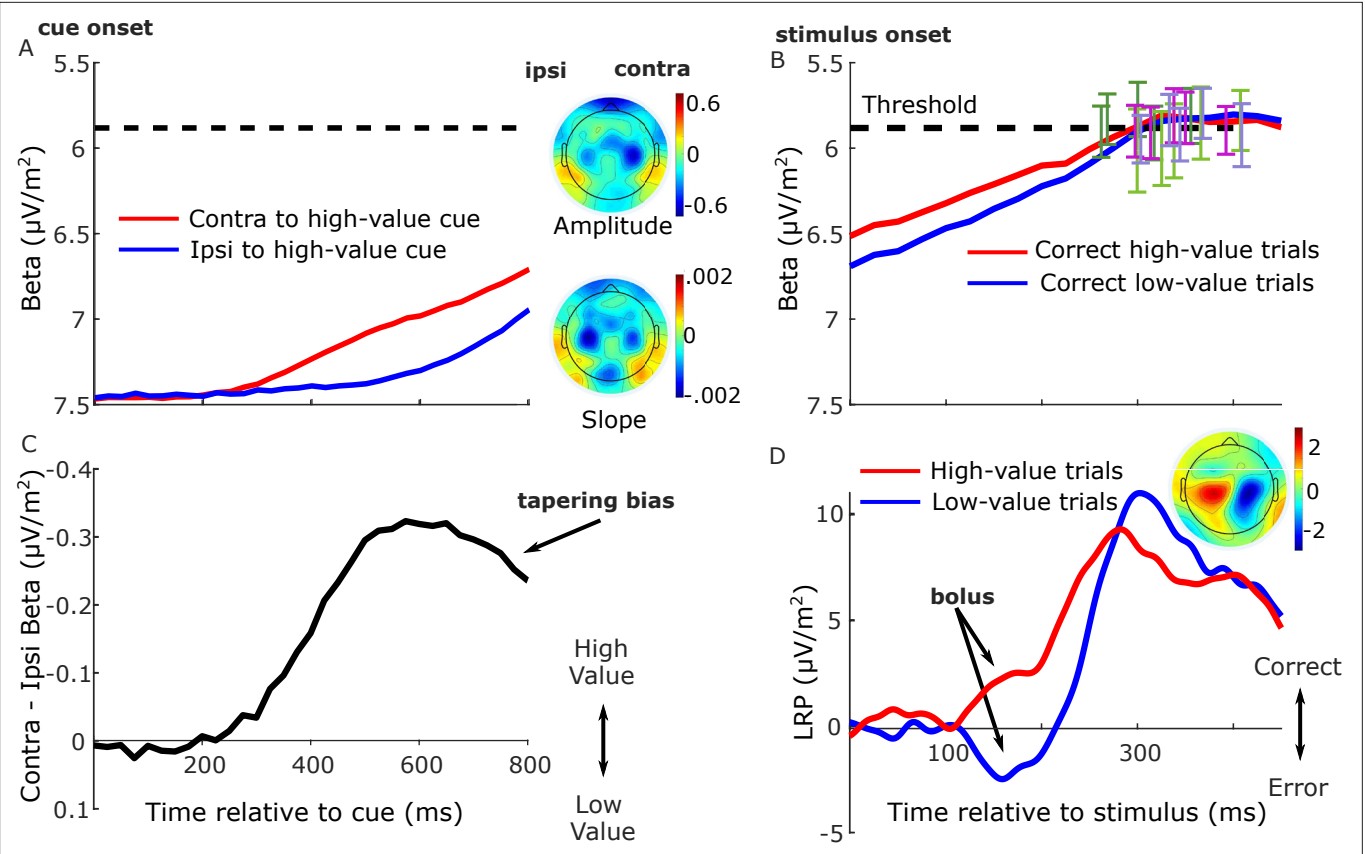

**Figure 2.** Grand-average (*n* = 17) electroencephalography (EEG) signatures of motor preparation. (**A**) Unilateral beta amplitude, contralateral to high- and low-value alternatives in the period after the cue and before the motion stimulus appeared at 850 or 900 ms. Note that the *Y*-axis is flipped such that decreasing amplitude (increasing motor preparation) is upwards. Topographies are for left-cued trials averaged with the right–left flipped topography for right-cued trials, so that the right side of the head plot represents the hemisphere contralateral to the high-value side. Amplitude topography reflects beta amplitude at 750 ms relative to amplitude at cue onset, and slope is measured from 700 to 800 ms. (**B**) Beta amplitude contralateral to response for correct trials only, relative to stimulus onset. Error bars are the standard errors of amplitudes 50 ms before response, with between-participant variability factored out, plotted against response time (RT). Trials were separated by session and coherence, showing high- and low-value correct trials median-split by RT and low-value error trials. (**C**) Relative motor preparation (the difference between the waveforms in panel A), highlighting the pre-stimulus decline due to steeper low-value urgency. (**D**) Lateralized readiness potential (LRP): ipsilateral–contralateral to correct response calculated at standard sites C3/C4, so that deflection upward corresponds to relative motor preparation in the correct direction. LRP waveforms were baseline corrected with respect to the interval −50 to 50 ms to focus on local stimulus-evoked dynamics. Topography shows the difference in amplitude between left- and right-cued trials at 150–180 ms relative to baseline. All waveforms derived from all trials regardless of accuracy unless otherwise stated.

The online version of this article includes the following figure supplement(s) for figure 2:

**Figure supplement 1.** Slope of each individual's beta decrease at 700–800 ms after the cue, for motor preparation contralateral to the high- and low-value alternatives, averaged across regimes.

**Figure supplement 2.** A slow-moving posterior potential interfered with measurement of the lateralized readiness potential (LRP) between cue and motion stimulus, leading us to rely solely on beta-band activity to examine anticipatory motor preparation.

consequence of these differences in onset and buildup rate, the bias in relative motor preparation favoring the high-value cue peaked at around 600 ms post-cue and then began to decline before stimulus onset (*Figure 2C*).

Next, to trace the rapid stimulus-evoked dynamics of the decision process with higher temporal resolution, we examined the broadband lateralized readiness potential (LRP). This differential signal represents the relative motor preparation dynamics between the hands associated with the correct and error responses (*Afacan-Seref et al., 2018*; *Gluth et al., 2013*; *Gratton et al., 1988*; *Noorbaloochi et al., 2015*; *van Vugt et al., 2014*), here examined relative to a peri-stimulus baseline interval (−50 to 50 ms) in order to emphasize fast stimulus-evoked dynamics (*Figure 2D*; see also

*Figure 2—figure supplement 2* for an analysis of the pre-stimulus LRP). Beginning approximately 100 ms after the stimulus, there was a deflection in the direction of the cued choice (in the correct direction for high-value trials and incorrect direction for low-value trials, $F(1,16) = 20.3$, p < 0.001, partial $\eta^2 = 0.56$, effect of value on the mean LRP from 150 to 180 ms, *Figure 2D*). We refer to this initial deflection as a 'bolus', following a similar finding by *Noorbaloochi et al., 2015*. The sensory evidence appears to begin to affect motor preparation at around 150 ms when the LRP for the low-value trials begins to turn around and build in the correct direction.

Together these signals indicate that motor preparation passed through several key phases. Anticipatory buildup began first for the high-value alternative, followed by low-value preparation which, beginning to compensate for its lower level, reached a higher buildup rate before stimulus onset, constituting a negative buildup-rate bias. Then, stimulus onset evoked a brief value-biased deflection, consistent with a positive drift-rate bias effect, before giving way to a final phase dominated by discriminatory sensory information.

## Model development

We next sought to construct a decision process model that can capture both behavior and the motor preparation dynamics described above. Probably the most widely used evidence accumulation model for two alternative decision making is the diffusion decision model (DDM, *Ratcliff, 1978*), which describes a one-dimensional stationary evidence-accumulation process beginning somewhere between two decision bounds and ending when one of the bounds is crossed, triggering the associated response action. The time this process takes is known as the *decision time*, which is added to a *nondecision time* (accounting for sensory encoding and motor execution times) to produce the final RT. This model has been successfully fit to the quantiles of RT distributions (e.g., *Figure 1—figure supplement 1*) for correct and error responses across a wide range of perceptual decision contexts. Traditionally, value biases can be incorporated into this framework by either biasing the starting point closer to one bound than the other or biasing the rate of evidence accumulation, the former of which generally better describes behavior (*Ratcliff and McKoon, 2008*). However, researchers have found that when there is a need to respond quickly, a stationary evidence-accumulation model is not sufficient to capture the pattern of value biases in behavior, which exhibits a dynamic transition from early, value-driven responses to later evidence-based ones. Accounting for this fast value-biased behavior in a DDM framework has instead required a nonstationary drift rate; either a dual-phase model with an initial value-based drift-rate transitioning to a later evidence-based one (*Diederich and Busemeyer, 2006*), or combining a constant drift-rate bias with a gradually increasing sensory evidence function (*Afacan-Seref et al., 2018*). Alternatively, *Noorbaloochi et al., 2015* proposed a linear ballistic accumulator model with a probabilistic fast guess component that was driven by the value information. However, in each of these approaches evidence accumulation begins from a stable starting point, meaning they could not account for the dynamic biased anticipatory motor preparation activity.

### Combined urgency + evidence accumulation model

As noted above, we interpreted the anticipatory beta changes to be reflective of a dynamic urgency driving the motor preparation for each alternative toward its threshold, independent of sensory evidence. Urgency has been found to be necessary to explain the more symmetrical RT distributions found in many speed-pressured tasks, as well as the sometimes-strong decline in accuracy for longer RTs in these conditions. Urgency has been implemented computationally in a variety of ways, reviewed in detail by *Smith and Ratcliff, 2021* and *Trueblood et al., 2021*. While models assuming little or no accumulation over time characterize urgency as a 'gain' function that multiplies the momentary evidence, models centered on evidence accumulation assume that urgency adds to cumulative evidence in a DV with a fixed threshold, which is mathematically equivalent to a bound on cumulative evidence that collapses over time (*Drugowitsch et al., 2012*; *Evans et al., 2020*; *Hawkins et al., 2015*; *Malhotra et al., 2017*). The latter, additive urgency implementation is consistent with neurophysiological signatures of urgency found across multiple evidence strengths including zero-mean evidence (*Churchland et al., 2008*; *Hanks et al., 2011*) and provides the most natural interpretation of the beta signals here due to their anticipatory, pre-stimulus buildup before evidence accumulation was possible. We therefore drew on a recently proposed model for decisions biased by prior expectations with two discrete levels: the one-dimensional accumulation of stimulus-evoked activity (noisy

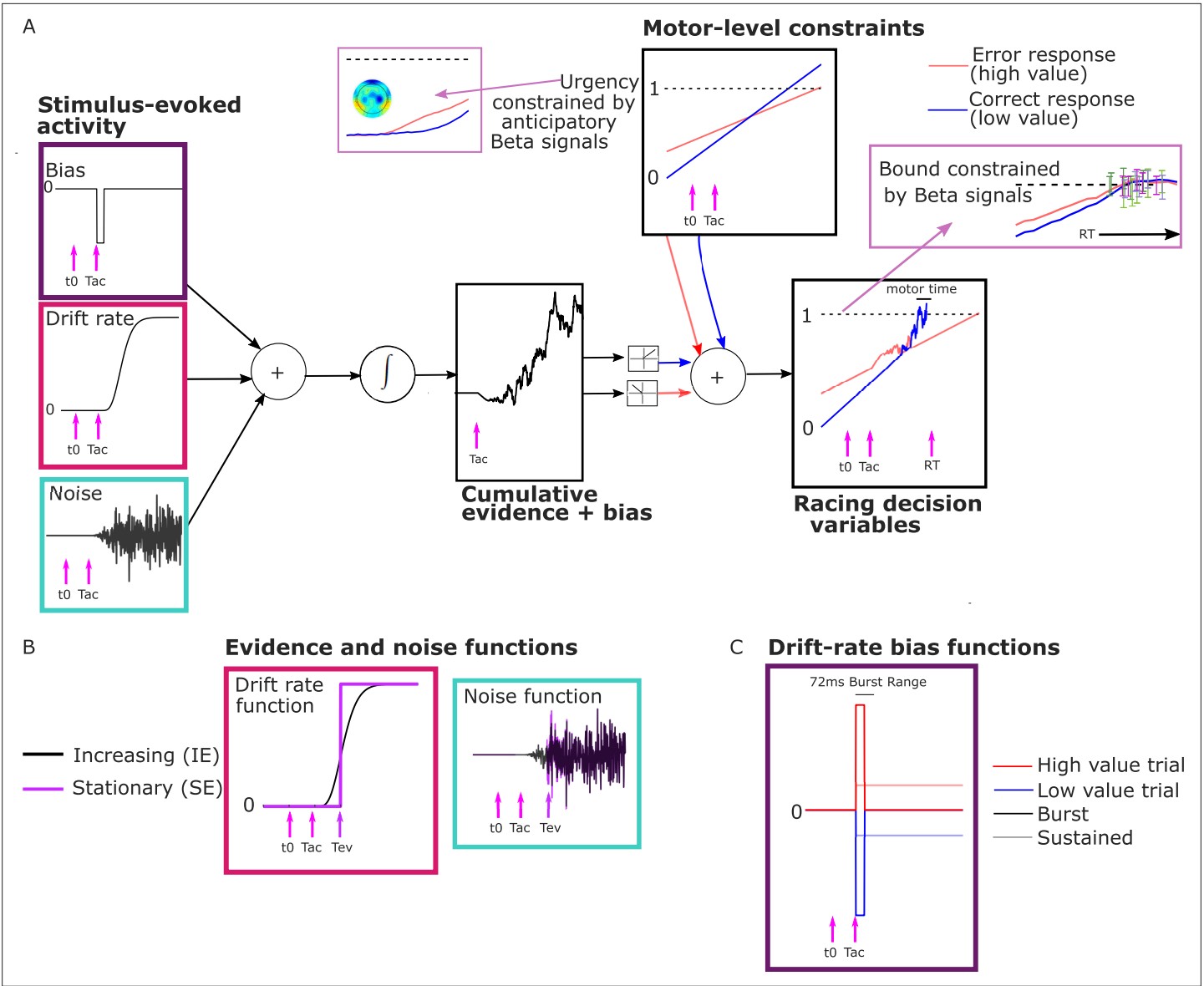

**Figure 3.** Model schematic. (**A**) Components of the model with a transient burst of stimulus-evoked bias and increasing evidence ('BurstIE'), with example traces for the cumulative sum of evidence plus bias, urgency and the resultant motor-level decision variable (DV) traces from a simulated low-value trial. A delay $T_{ac}$ after stimulus onset, $t0$, the combination of a sudden detection-triggered bias function and growing, noisy sensory evidence began to be accumulated, and with the addition of urgency drove the race between two DVs toward the threshold. The cumulative evidence plus bias was half-wave rectified such that (positive) evidence toward the correct (low-value) response was added to the low-value urgency signal, and vice versa. (**B**) Alternative evidence and noise functions. For stationary evidence (SE) models both stepped abruptly to their asymptotic value whereas for increasing evidence (IE) models both increased according to a gamma function. (**C**) Alternative drift-rate bias functions. For 'Burst' models the duration of bias was short, with a maximum of 72 ms, whereas sustained drift-rate bias ('Sust') models had a bias that continued throughout the trial. Waveforms are not drawn to scale.

sensory evidence and bias) is fed to a 'motor' level where it is combined additively with evidence-independent buildup components that linearly increase with time (*Murphy et al., 2016*; *Steinemann et al., 2018*) to generate the motor-level DVs engaging in a race to the bound (*Kelly et al., 2021*, *Figure 3A*).

Distinct pre-stimulus starting levels were set for the DV contralateral (parameter $Z_c$) and ipsilateral ($Z_i$) to the direction of the value cue for each regime. Extrapolating from the anticipatory motor preparation buildup, we assumed the operation of linearly increasing urgency, which was also biased by the value cue. The urgency buildup rates varied from trial to trial independently for the two response

alternatives, in a Gaussian distribution with means $U_{c,i}$ and standard deviation $s_u$. We assume in all models that the accumulation process takes an additive combination of noisy stimulus evidence plus a stimulus-evoked bias, both of which are implemented in alternative ways for comparison as detailed below. We refer to that combination as the 'cumulative evidence plus bias' function, $x(t)$. The DVs were then generated by adding the cumulative evidence plus bias in favor of either alternative to the corresponding motor-level urgency signal, triggering a decision at the 'decision time' when the first reached the bound:

$$DV_1(t) = m_1(t) + \lfloor x(t) \rfloor \tag{1}$$

$$DV_2(t) = m_2(t) + \lfloor -x(t) \rfloor \tag{2}$$

Here, $DV_1$ and $DV_2$ represent the DVs for the correct and incorrect responses, respectively, which were updated in our simulations at a time interval $dt = 1$ ms. $m_1$ and $m_2$ represent the motor-level urgency contributions contralateral and ipsilateral to the cued direction on high-value trials, and the reverse on low-value trials. The motor-level urgency contribution was defined as:

$$m_1(t) = z_1 + u_1 \cdot (t - T_z) \tag{3}$$

$$m_2(t) = z_2 + u_2 \cdot (t - T_z) \tag{4}$$

$z_1$ and $z_2$ represent the starting levels for the DVs at pre-stimulus time, $T_z$, at which the starting beta levels are measured; and $u_1$ and $u_2$ represent the urgency rates for the two alternatives on that trial. For example, in a high-value trial (in which the cued direction is the correct response):

$$u_1 \sim N(U_c, s_u); z_1 = Z_c, \tag{5}$$

$$u_2 \sim N(U_i, s_u); z_2 = Z_i \tag{6}$$

The cumulative evidence plus bias, $x(t)$ is positive in the direction of the correct response, and the half-wave rectification operation, $\lfloor x \rfloor = max(0, x)$, apportions the positive and negative components to the appropriate DVs. All the above equations are defined for the time following $T_z$.

The trial RT was obtained by adding to the decision time a *motor execution time*; this varied from trial to trial on a uniform distribution with mean $T_r$ which varied between the regimes, and range $s_t$. Allowing for regime differences in motor execution was important as its timing is known to be affected by speed/accuracy settings (*Kelly et al., 2021*; *Rinkenauer et al., 2004*; *Weindel et al., 2021*). In previous work, we had constrained the mean motor execution time parameter using an EEG motor-evoked potential (*Kelly et al., 2021*). However, likely due to the substantially increased model constraints in the current study (see Neural constraints), we found in preliminary analyses that constraining the motor execution times in this way was detrimental to our fits. The cumulative evidence plus bias function was initiated at the time of stimulus onset $x(0) = 0$, and updated according to the following equation:

$$x(t) = x(t - dt) + B(t) \cdot dt + \mu(t) \cdot dt + w(t) \cdot \sqrt{dt} \tag{7}$$

Here, $B(t)$ represents the stimulus-evoked bias and $\mu(t)$ is the drift rate of the evidence. The within-trial noise, $w(t)$, is Gaussian distributed with standard deviation $\sigma(t)$:

$$w(t) \sim N(0, \sigma(t)) \tag{8}$$

## Neural constraints

Based on the principle that neural constraints permit greater model complexity without unduly increasing degrees of freedom (*O'Connell et al., 2018*), from the anticipatory motor preparation signals we adopted constraints on not just starting levels (*Kelly et al., 2021*) but also the biased mean urgency buildup rates. The mean beta starting levels (750 ms post-cue) and slopes (from 700 to 800 ms post-cue) were calculated for each regime across participants. To obtain the model parameters, we linearly re-scaled the beta signals within a range from 0, corresponding to the lowest starting level, to a fixed bound of 1 corresponding to the beta threshold—the average value of beta contralateral to the chosen hand across all conditions 50 ms prior to response (see Figure 4A). The setting of the

**Table 1.** Electroencephalography (EEG)-constrained parameters.

| Parameter | Symbol | High coherence | Low coherence | Interleaved |
|---|---|---|---|---|
| Starting point contralateral to high value | $Z_c$ | 0.33 | 0.3 | 0.2 |
| Starting point ipsilateral to high value | $Z_i$ | 0.14 | 0.003 | 0 |
| Mean urgency rate contralateral to high value | $U_c$ | 1.33 | 1.06 | 1.26 |
| Mean urgency rate ipsilateral to high value | $U_i$ | 1.78 | 1.66 | 1.76 |

bound at 1 was an arbitrary choice and serves as the scaling parameter for the model. The starting levels and mean rates of urgency buildup for the high- and low-value alternatives were set to equal the amplitude and temporal slope of the corresponding scaled beta signals for each regime (**Table 1**).

Within this neurally constrained urgency model framework, we fit several alternative bounded accumulation models to the data for comparison. Specifically, we explored whether the data were better captured by a stationary (**Ratcliff and McKoon, 2008**) or growing (**Afacan-Seref et al., 2018**; **Smith and Lilburn, 2020**) evidence function, and by a sustained (**Afacan-Seref et al., 2018**) or transient (**Diederich and Busemeyer, 2006**) drift-rate bias, by comparing four main model variants that featured two plausible alternative ways to implement noisy evidence accumulation and two different stimulus-evoked biasing mechanisms.

## Evidence and noise functions

We compared models with a standard *stationary evidence* (SE) function with abrupt onset to *increasing evidence* (IE) models where the evidence and noise gradually grow with time (**Smith et al., 2014**; **Smith and Lilburn, 2020**; **Figure 3B**). Both model types had an asymptotic drift-rate parameter, $v$, to which the mean of the sensory evidence stepped (SE) or gradually tended (IE), for each coherence level. A single within-trial noise parameter ($s$) dictated the asymptotic standard deviation of Gaussian-distributed within-trial noise. We also estimated an onset time for accumulation, $T_{ac}$, relative to stimulus onset. In the SE models, this parameter signaled the onset of the bias accumulation (see below), while the noisy evidence stepped up at a later time, $T_{ev}$:

$$\mu_{SE}\left(t\right) = \begin{cases} v \; if \, t > T_{ev} \\ 0 \, otherwise \end{cases} \tag{9}$$

$$\sigma_{SE}\left(t\right) = \begin{cases} s \; if \, t > T_{ev} \\ 0 \, otherwise \end{cases} \tag{10}$$

In the IE models, the bias, evidence, and noise functions all began at $T_{ac}$. The IE and noise functions used were those developed for a time-changed diffusion model (**Smith et al., 2014**; **Smith and Lilburn, 2020**) in which the drift rate $v$, and diffusion coefficient $s^2$ (the squared standard deviation of the Gaussian-distributed within-trial noise), are both scaled by a growth rate function $\vartheta$:

$$\mu_{IE}\left(t\right) = v.\vartheta\left(t\right) \tag{11}$$

$$\sigma_{IE}\left(t\right) = s.\sqrt{\vartheta\left(t\right)} \tag{12}$$

Following (**Smith and Lilburn, 2020**, see equation 9), $\vartheta$ took the form of an incomplete gamma function with rate $\beta$, where the argument $n$ and $\beta$ were free parameters:

$$\vartheta\left(t\right) = \begin{cases} \frac{1}{\Gamma(n)}\int_0^{\beta(t-T_{ac})} e^{-r}r^{n-1}dr, \, if \, t > T_{ac} \\ 0, otherwise \end{cases} \tag{13}$$

In this equation, $\Gamma\left(n\right)$ is the gamma function. The shape of the function obtained by one of our model fits is shown in **Figure 3B**.

## Stimulus-evoked bias functions

We also compared two alternative implementations of a drift-rate bias across different model variants. One featured a sustained drift-rate bias ('Sust') which began at $T_{ac}$ and lasted until response. The other featured a shorter transient bias, inspired by the apparent concentrated burst of value-biased activity ('Burst') before evidence accumulation took hold in the LRP (*Figure 3C*). Both functions involved a bias magnitude parameter ($\nu_b$) for each regime:

$$B_{Sust}(t) = \begin{cases} \pm\nu_b, & if\ t \geq T_{ac} \\ 0, & otherwise \end{cases} \tag{14}$$

$$B_{Burst}(t) = \begin{cases} \pm\nu_b, & if\ T_{ac} \leq t \leq (T_{ac} + BurstT) \\ 0, & otherwise \end{cases} \tag{15}$$

The bias factor $\pm\nu_b$ was positive for high-value trials and negative for low-value trials. The 'Burst' was composed of a drift-rate bias beginning at $T_{ac}$ whose duration $BurstT$ varied on a uniform distribution from 0 to 72 ms. In preliminary analyses, we found that the burst magnitude and its range of durations could trade off each other such that equivalent fits to behavior could be found for a wide range of values of the latter. We thus fixed the maximum duration to 72 ms because it produced a simulated-DV bolus similar in duration to the real LRP (*Figure 4B, C*; see Methods). We also restricted $T_{ac}$ to a narrow range of 90–100 ms in the fits, close to the apparent onset of the real LRP bolus; we did not find that expanding this range helped the models to converge.

## Model fits

Models were fit to the group average of the RT quantiles (see Methods). We did not fit the models to individual participants because, in contrast to models solely fit to behavior where each individual's data can be taken as an accurate reflection of the outcomes of their true individual decision process, our neurally constrained models constrain certain key parameters to equal EEG beta-amplitude measures. These EEG measures are much less reliable on an individual-participant level, where it is not unusual to have certain individuals showing no signal at all due to factors independent of the decision process such as brain geometry. We therefore conduct the modeling on a grand-average level because grand-average beta-amplitude dynamics are much more robust.

The increasing-evidence (IE) models performed better than the stationary-evidence (SE) models, with the BurstIE model providing the best fit to behavior (*Table 2*). This model captured all the main qualitative features of the RT distributions, including the indistinguishable (value-driven) leading edges of correct high-value and incorrect low-value trials (*Figure 4D, E*), and the transition from value- to evidence-based responses visible in the low-value conditional accuracy functions (CAFs, *Figure 4F*). Although the SustIE, BurstSE, and SustSE models exhibited a less close quantitative fit to behavior as reflected in Akaike's information criterion (AIC) and Akaike weights (*W*), qualitatively, they all captured the main behavioral patterns reasonably well including the biased fast guess responses (*Figure 4—figure supplements 1–3*). The estimated parameters for these four primary models are given in *Table 3*.

We tested four additional versions of the IE model to assess the contribution of the constrained urgency and stimulus-evoked bias to the fits (*Table 2*). First, allowing the urgency rates to be free parameters, but unbiased by value (*Kelly et al., 2021*), did not capture the behavior as well as the constrained BurstIE model. Then, a model with constrained urgency but no stimulus-evoked bias produced a far inferior fit. These results suggest that in addition to accounting for the slow temporal integration properties of sensory evidence encoding, incorporating both key insights gained from the EEG signals was important in capturing behavior. We then verified the specific contribution of quantitative differences across regimes in the urgency effects measured in the beta signals by showing that swapping the neural constraints across regimes substantially worsened the fit.

To serve as a benchmark, we also fit the DDM, with stationary evidence accumulation and allowing for bias only in the starting point of evidence accumulation (*Ratcliff and McKoon, 2008*), whose performance was markedly worse than all of the neurally constrained alternatives. Indeed, this poor performance was expected given there are substantive qualitative features of the data that the DDM

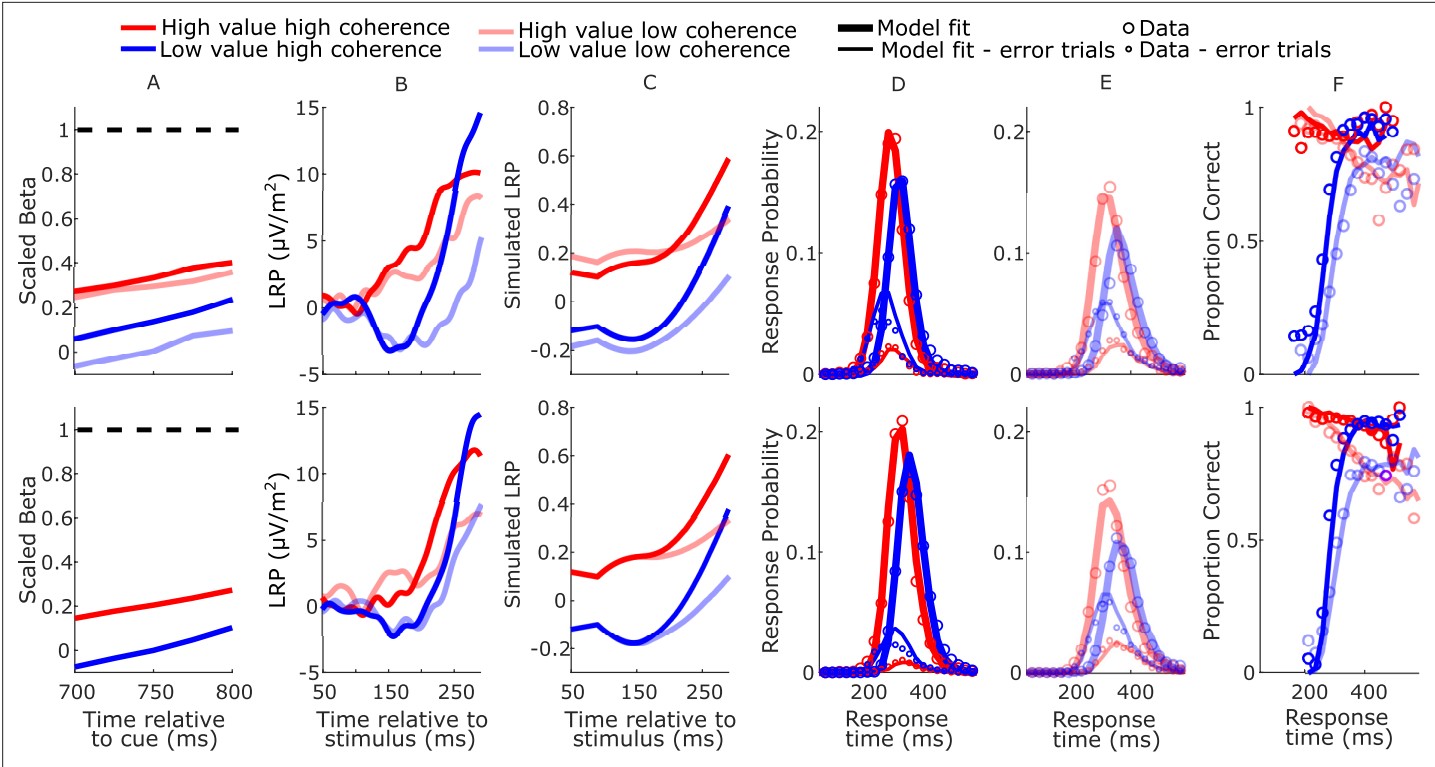

**Figure 4.** Real and model-simulated waveforms and behavior for blocked session (top row) and interleaved session (bottom row). (**A**) Scaled beta signals used to constrain the models. The high- versus low-value difference in starting level varied across regime (Regime × Value interaction $F_{(2,32)}$ = 4.1, p = 0.03, partial $\eta^2$ = 0.84; pairwise comparisons of value-difference indicated low-coherence blocked > high-coherence blocked, p = 0.01). The Regime × Value interaction for slope was not statistically significant ($F_{(2,32)}$ = 0.11, p = 0.89, partial $\eta^2$ = 0.96); (**B**) real lateralized readiness potential (LRP). There was a significant interaction in bolus amplitude (mean LRP from 150 to 180 ms) between Value and Condition ($F_{(3,48)}$ = 2.9, p = 0.04, partial $\eta^2$ = 0.16). Pairwise comparisons of the value difference provided moderate evidence that the blocked high-coherence condition had a larger difference than the interleaved high-coherence condition (p = 0.09, BF10 = 3.87); there were no significant differences between the other conditions (all p > 0.23). (**C**) Mean simulated trajectories of the difference between correct and incorrect decision variables (DVs) from the best-fitting model with burst drift-rate bias and increasing evidence (BurstIE); (**D, E**) real (circles) and model-simulated (solid lines) response time (RT) distributions. (**F**) Real and model-simulated conditional accuracy functions (CAFs). All waveforms derived from all trials regardless of accuracy.

The online version of this article includes the following figure supplement(s) for figure 4:

**Figure supplement 1.** SustIE model-simulated waveforms and behavior for blocked session (top row) and interleaved session (bottom row).

**Figure supplement 2.** BurstSE model-simulated waveforms and behavior for blocked session (top row) and interleaved session (bottom row).

**Figure supplement 3.** SustSE model-simulated waveforms and behavior for blocked session (top row) and interleaved session (bottom row).

**Figure supplement 4.** Diffusion decision model (DDM) model-simulated waveforms and behavior for blocked session (top row) and interleaved session (bottom row).

**Figure supplement 5.** Results of Jackknifing analysis.

is not equipped to capture, such as the value-driven leading edge, the fast transition from value- to evidence-based responses, and the symmetric RT distributions as has been established before (*Afacan-Seref et al., 2018*; *Diederich and Busemeyer, 2006*; *Kelly et al., 2021*). The estimated parameters for the DDM are given in *Table 4*.

Finally, in the last seven rows of *Table 2* we report the performance of selected neurally-constrained models that incorporate additional parameters which were included in a neurally-constrained model from previous work (*Kelly et al., 2021*) but had little effect here. First, a central finding from *Kelly et al., 2021*, which involved an extreme speed-pressure manipulation, was that the drift-rate parameter increased under speed pressure for the same stimulus coherence. Thus, the: 'BurstIE + drift boost' model allowed an additional drift boost parameter in the high-coherence blocked condition, relative to the high-coherence interleaved condition. This resulted in an almost identical $G^2$, suggesting that in this case the much more subtle speed-pressure manipulation between the conditions was not

**Table 2.** Goodness-of-fit metrics.

| Model | Stimulus-evoked bias | Evidence | $k$ | $G^2$ | AIC | $W$ |
|---|---|---|---|---|---|---|
| BurstIE | Burst | Increasing | 14 | 43 | 71 | 0.47 |
| SustIE | Sustained | Increasing | 14 | 60 | 88 | 0.0001 |
| BurstSE | Burst | Stationary | 13 | 69 | 95 | 0 |
| SustSE | Sustained | Stationary | 13 | 89 | 115 | 0 |
| Unbiased urgency slopes | Burst | Increasing | 17 | 54 | 88 | 0.0001 |
| Urgency-only bias | None | Increasing | 11 | 362 | 384 | 0 |
| DDM | None | Stationary | 14 | 606 | 634 | 0 |
| Constraints-Swap 1 | Burst | Increasing | 14 | 272 | 300 | 0 |
| Constraints-Swap 2 | Burst | Increasing | 14 | 122 | 150 | 0 |
| BurstIE + drift boost | Burst | Increasing | 15 | 42 | 72 | 0.22 |
| BurstIE + $s_z$ | Burst | Increasing | 15 | 42 | 72 | 0.31 |
| SustIE + $s_z$ | Sustained | Increasing | 15 | 59 | 89 | 0 |
| BurstSE + $s_z$ | Burst | Stationary | 14 | 69 | 97 | 0 |
| SustSE + $s_z$ | Sustained | Stationary | 14 | 92 | 120 | 0 |
| BurstSE + $sT_{ev}$ | Burst | Stationary | 14 | 64 | 92 | 0 |
| SustSE + $sT_{ev}$ | Sustained | Stationary | 14 | 92 | 120 | 0 |

Goodness-of-fit quantified by chi-squared statistic, $G^2$. Model comparison was performed using AIC, which penalises for the number of free parameters ($k$). The Akaike weights ($W$) shown, which can be cautiously interpreted as the probability that each model is the best in the set, are calculated here based on the set of models in this table. The probability mass is shared between the different versions of the BurstIE model. In the two Constraints-Swap models, the constrained parameters for (A) high- coherence, (B) low- coherence, and (C) interleaved blocks were taken from the neural signals corresponding to [B,C,A] (Swap 1) and [C,A,B] (Swap 2), respectively.

sufficient to replicate the effect. Second, their model had a uniformly distributed starting level variability with a range parameter, $s_z$, applied independently to the constrained mean starting levels of the DVs. This parameter did not improve our fits to any of the four neurally constrained models (listed in rows 11–14). Third, it was possible that the effect of the gradual integration of motion evidence could be captured in the SE models by allowing for variability in the evidence onset time, $T_{ev}$. Whereas *Kelly et al., 2021* incorporated variability in accumulation onset relative to a fixed evidence onset time, it was more convenient here to incorporate a qualitatively similar feature by varying evidence onset, since accumulation onset was anchored to the onset of the LRP bolus response. We found that adding such variability, uniformly distributed with range $sTev$, very slightly improved performance of the BurstSE model and did not help the SustSE model. Neither were improved to an extent where they could compete with the best-fitting BurstIE model.

## DV simulations

We qualitatively explored the correspondence between the fast neural dynamics of the LRP and simulated decision process by plotting the difference between the two DVs (*Figure 4B, C*). The starting levels are not comparable because, unlike the simulated process, the real LRP was baseline corrected, and the initially decreasing value bias in the simulated waveforms is not seen in the LRP due to interfering posterior slow potentials (see *Figure 2—figure supplement 2*). There was, however, good correspondence between the dynamics from the onset of the deflection, which was notably absent in the alternative SustIE and SustSE model simulations (*Figure 4—figure supplements 1 and 3*). The BurstIE model effectively captured aspects of both EEG motor preparation signatures through its distinct countervailing biasing mechanisms. While our previous work compared the absolute value of the simulated cumulative evidence and bias function ($x(t)$) to the centroparietal positivity—an event-related potential (ERP) thought to be related to evidence accumulation (*Kelly et al., 2021*)—here this

**Table 3.** Estimated parameters for the four main models.

| Parameter | Symbol | BurstIE | SustIE | BurstSE | SustSE |
|---|---|---|---|---|---|
| Asymptotic drift rate (high coherence) | $v_h$ | 6.4 | 8.4 | 4.9 | 4.6 |
| Asymptotic drift rate (low coherence) | $v_l$ | 2.8 | 3.3 | 2.1 | 2.1 |
| Drift-rate bias (high-coherence blocked) | $v_{bh}$ | 2.4 | 0.63 | 2.3 | 0.51 |
| Drift-rate bias (low-coherence blocked) | $v_{bl}$ | 2.3 | 0.49 | 2.4 | 0.46 |
| Drift-rate bias (interleaved) | $v_{bi}$ | 3.1 | 0.74 | 3.1 | 0.63 |
| Within-trial noise asymptotic standard deviation | $s$ | 1.13 | 1.16 | 0.93 | 0.81 |
| Accumulation onset time (ms) | $T_{ac}$ | 90 | 90 | 91 | 91 |
| Burst duration range (ms) | $b_{range}$ | **72** | — | **72** | — |
| $\vartheta\left(t\right)$ — rate | $\beta$ | 54.9 | 41.4 | — | — |
| $\vartheta\left(t\right)$ — argument | $n$ | 6.9 | 6.7 | — | — |
| Evidence onset time (ms) | $T_{ev}$ | — | — | 205 | 223 |
| Mean motor time (high-coherence blocked) (ms) | $T_{rh}$ | 86 | 72 | 73 | 57 |
| Mean motor time (low-coherence blocked) (ms) | $T_{rl}$ | 85 | 67 | 74 | 54 |
| Mean motor time (interleaved) (ms) | $T_{ri}$ | 95 | 79 | 84 | 63 |
| Urgency-rate variability | $s_u$ | 0.42 | 0.46 | 0.39 | 0.4 |
| Motor time variability (ms) | $s_t$ | 65 | 65 | 81 | 80 |

Note: Fixed parameter shown in bold typeface.

**Table 4.** Estimated parameters for the diffusion decision model (DDM).

| Parameter | Symbol | DDM |
|---|---|---|
| Drift rate (high coherence) | $v_h$ | 6.34 |
| Drift rate (low coherence) | $v_l$ | 3.5 |
| Bound (high-coherence blocked) | $a_h$ | 0.17 |
| Bound (low-coherence blocked) | $a_l$ | 0.18 |
| Bound (interleaved) | $a_i$ | 0.16 |
| Starting point bias (high-coherence blocked) | $z_{b\_h}$ | 0.12 |
| Starting point bias (low-coherence blocked) | $z_{b\_l}$ | 0.1 |
| Starting point bias (interleaved) | $z_{b\_i}$ | 0.11 |
| Nondecision time (high-coherence blocked) (ms) | $T_{er\_h}$ | 0.27 |
| Nondecision time (low-coherence blocked) (ms) | $T_{er\_l}$ | 0.3 |
| Nondecision time (interleaved) (ms) | $T_{er\_i}$ | 0.31 |
| Starting point variability | $s_z$ | 0.09 |
| Nondecision time variability | $s_t$ | 0.13 |
| Drift-rate variability | $\eta$ | 6.39 |

component was obscured by large potentials evoked by the sudden stimulus onset, and thus could not be reliably used in the same way.

## Jackknifing procedure for model comparison

The variability in individual-participant EEG precluded us from performing neurally constrained modeling at the individual level, so it was not possible to verify that this model comparison would hold for all participants. While the analysis represented in *Figure 1—figure supplements 1 and 2* reassured us that the quantile-averaging of the data did not cause distortion, we nevertheless sought to take a step toward quantifying how much our participant selection affected the model comparison results. To this end, we repeated the model comparison for the 4 main neurallyconstrained models and the DDM 17 times in turn with one participant excluded each time. The BurstIE model was strongly preferred for all of the samples (see *Figure 4—figure supplement 5*).

## Discussion

Convergent evidence from motor preparation signals and behavioral modeling demonstrated that a dynamic sequence of opposing value biases and nonstationary evidence accumulation all played important roles in forming the rapid, multiphasic decisions on this task. In most decision-making models a 'starting point bias' parameter—shifting the starting point of accumulation—treats anticipatory biases as static adjustments before the process begins (*Leite and Ratcliff, 2011*; *Mulder et al., 2012*). Here, far from creating a stable starting point to kick off a stationary decision process, we found a dynamic pattern of biased motor preparation that is best understood as a two-dimensional race beginning well in advance of the stimulus. Constraining a behavioral model with these signals enabled us to characterize a surprisingly complex process, revealing biasing mechanisms that would otherwise have been inaccessible.

In agreement with previous research that has called for nonstationary accounts of value biasing in time-pressured decisions (*Diederich and Busemeyer, 2006*), we found that the value bias was largely concentrated in the early part of the process. The particular dynamics of the RDK stimulus, featuring a substantial lag between stimulus onset and the emergence of discriminatory sensory evidence, may have provided a focal point for the bias to be expressed separately from the evidence itself; indeed the model comparison very clearly favored the growing sensory evidence and noise. However, the signature expressions of this sequential detection–discrimination effect—namely, the almost purely value-driven nature of both the leading edge of RT distributions and of the initial stimulus-evoked LRP deflection—are observed also for discriminations of stimulus displacement (*Noorbaloochi et al., 2015*) and color (*Afacan-Seref et al., 2018*), suggesting the phenomenon generalizes beyond the RDK stimulus. While our findings indicate that a strong transient drift-rate bias better captures the data relative to a sustained, constant bias, the possibility of a hybrid of the two, where the initial detection-triggered burst reduces to a smaller sustained bias, was not tested because it was assumed to go beyond a reasonable number of free parameters. Thus, uncertainty remains regarding the exact temporal profile of this stimulus-evoked bias, and we cannot say that it fully disappears beyond the burst.

The dynamic shift from value to evidence-driven accumulation is reminiscent of conflict tasks, for which a stationary drift rate is similarly insufficient to describe the observed behavioral patterns. In these tasks, the context in which a perceptual stimulus is presented (i.e., features of the stimulus that are irrelevant to the task requirements) can be congruent with either the correct or the incorrect response. The latter case causes conflict that results in slower and more error-prone responding (*Eriksen and Eriksen, 1974*; *Lu and Proctor, 1995*; *MacLeod, 1991*), and produces signatures of competing motor plans in the LRP that are similar to those found here (*Gratton et al., 1988*). Prominent accounts of these tasks posit that an automatic processing of the stimulus happens in parallel with the controlled (decision) process (*Servant et al., 2016*; *Ulrich et al., 2015*). It is plausible that the LRP 'bolus' in our study could arise from a related mechanism in which the value cue automatically 'primes' a response, although it seems likely that value-biased responding is more intentional since it may confer a benefit in terms of the increased reward. Indeed, the patterns of biased anticipatory motor preparation we see in this study cannot be present in tasks where the conflict does not arise until after stimulus onset; in such tasks the anticipatory mu/beta buildup activity while present

is unbiased (*Feuerriegel et al., 2021*). In the case of these beta signals, the fact that the buildup happens earlier under speed pressure (*Kelly et al., 2021*) suggests that they are much more likely to be strategic rather than automatic, and we would not expect a bottom-up lateralization in response to the physical appearance of the cues due to their symmetric design. Nonetheless, even if different in nature, some of the functional dynamics arising from our value bias cues are interestingly similar to those arising from conflict tasks where both competing influences are externally presented.

The implication of a negative buildup-rate bias in urgency is counterintuitive but not completely without precedent. In the context of the DDM with unequal prior probabilities, *Moran, 2015* found that a negative drift-rate bias featured alongside a starting point bias in the optimal decision strategy under certain assumed bound settings, albeit not when bound settings were assumed controllable as part of the optimization calculation. Here, a similar tradeoff between the positive starting level bias and negative urgency-rate bias may have arisen from the fact that the greater the starting point bias, the greater the need for a steeper low-value urgency signal to give it a chance to overtake the high-value signal when the low-value DV represents the correct response.

Understanding the processes generating the behaviors in this task rested on the neurophysiological identification of strategic urgency biases. The anticipatory nature of the early beta signal buildup aided in specifically linking it to evidence-independent urgency, and its incorporation in the model was key to understanding the subsequent processing of the motion stimulus. We conducted the modeling here on a grand-average level because grand-average beta-amplitude dynamics are much more robust than those of individuals, but this meant that we were unable to examine individual differences in behavior. The extent to which these different forms of bias might trade off each other at the individual level remains for now an open question. Nevertheless, the finding of a negative urgency-rate bias as part of the participants' dominant strategy highlights the broad range of dynamic adjustments that can be made during fast-paced sensorimotor decisions.

## Methods

### Participants

The experiment involved one psychophysical training session and two EEG recording sessions. As the task was challenging, the training session served a dual purpose of giving participants the time to learn the task and to screen out those who found it too difficult. Twenty-nine adult human participants performed the training session. Eleven discontinued due to either insufficient performance on the task, or conflicting commitments. Eighteen participants (eight female) thus completed the two EEG sessions. Motor preparation biasing effects tend to be consistent and robust (e.g., effect sizes of at least $d$ = 1 for similar 'bolus' effects in *Afacan-Seref et al., 2018*), and 15–18 participants provide 80% power to detect medium-to-large effect sizes. Participants all had normal or corrected-to-normal vision. They each provided informed, written consent to the procedures, which were approved by the Ethics Committee of the School of Psychology at Trinity College Dublin, and the Human Research Ethics Committee for the Sciences, at University College Dublin. Participants were compensated with €20 for the training session and €32 for their participation in each EEG session with the potential to earn up to €12 extra depending on their performance. One of the participants was an author and the remainder were naive.

### Setup

Participants were seated in a dark booth, with their heads stabilized in a chin rest placed 57 cm from a cathode ray tube monitor (frame rate 75 Hz, resolution 1024 × 768) with a black background. They rested their left/right thumbs on the left/right buttons of a symmetric computer mouse secured to the table in front of them.

### Task

The task was programmed in Psychtoolbox for MATLAB (*Brainard, 1997*). Trials began with the presentation of a central gray 0.25° fixation square. Upon achieving fixation (4° radius detection window, EyeLink 1000, SR Research), a value cue replaced the fixation square after either 400 or 450 ms (randomly selected) and remained on screen, until the end of the trial (*Figure 1*). The cue consisted of equiluminant green and cyan arrows placed and pointing to the left and right of center, indicating

the directions that would be worth 30 points (high value) or 10 points (low value) if subsequently presented and correctly responded to with the corresponding hand within the deadline. Incorrect or late responses were worth 0 points. Color-value assignment was randomly counterbalanced across participants. The RDK stimulus (5° diameter) appeared and commenced moving either 850 or 900 ms (randomly selected) after cue onset and lasted 600 or 781 ms for the shorter or longer deadline conditions, respectively. Participants were required to maintain fixation throughout, and upon stimulus offset received feedback on whether they were 'Correct!', 'WRONG!', 'TOO SLOW!' or 'TOO EARLY! WAIT FOR CUE …' or 'WAYYY TOO SLOW!' if they did not respond at all before the dots turned off.

The task was performed in three blocked regimes: high coherence (51.2%) with a short deadline (365 ms); low coherence (19.2%) with a slightly longer deadline (475 ms); and interleaved high and low coherence with the longer deadline. The RDK stimulus was adapted from code from the Shadlen laboratory (*Gold and Shadlen, 2003*; *Roitman and Shadlen, 2002*). A set of white dots were presented within a circular aperture of 5° in diameter that was the same black color as the background. The dot density was 16.7 dots per °/s. One-third of the total number of dots was visible on screen at any one time; each dot remained on screen for one 13.3-ms frame and was replotted two frames later as the three sets of dots were alternated. Depending on the coherence level, each dot had either a 19.2% or 51.2% chance of being replotted by an offset in the direction of coherent motion at a rate of 5°/s. Otherwise the dots were randomly relocated within the aperture. The first onset of coherent motion thus occurred 40 ms (three frames) after the onset of the stimulus. If an offset dot was set to be plotted outside of the aperture, it was replotted in a random location on the edge of the aperture opposite to the direction of motion.

## Procedure

So that participants could become familiar with the task, and particularly get used to its fast pace, they performed one session of psychophysical training before the main experimental sessions. Blocks in the training sessions comprised 80 trials. The session began with blocks of high-coherence trials with a long deadline and without value bias (20 points for each direction; both arrow cues were yellow). The deadline was gradually reduced to 365 ms. The same procedure was then followed for low-coherence blocks. If participants had great difficulty with the low coherence, the experimenter gave them some further practice starting at 45% and gradually brought it down to 19.2%. Finally, participants practiced an equal number of biased blocks in the high-coherence, low-coherence, and interleaved high- and low-coherence regimes.

Participants performed the two blocked regimes (5 or 6 blocks each of 120 trials) in one EEG recording session and the interleaved regime (10 or 12 blocks) in the other. Due to experimenter error, one participant performed the blocked experimental session twice and we included the data from both sessions in our analyses. The blocks within each regime were run consecutively to ensure that participants would settle into a strategy, and the order of regimes and sessions was randomized. In training and throughout the EEG recording sessions, participants were encouraged to adopt a strategy that would maximize their points and were informed that the points earned in two randomly selected blocks (one per regime in the blocked session) would determine their bonus payment in each recording session. Participants were provided with the total number of points earned at the end of the block as well as the number of points missed in the block for each trial type (blue and green) to motivate them and help them determine whether they were biasing too much or too little. The experimenters helped participants interpret this feedback and when needed provided frequent reminders that it was important to pay attention to both the value cue and the stimulus and that there were no points awarded for late responses.

## Behavioral analyses

RTs were measured relative to the onset of the RDK stimulus. RTs less than 50 ms (0.23% of trials) were excluded from analyses and model fitting. Responses up to and beyond the deadline were included in all analyses so long as they occurred before the end of the RDK stimulus; trials without a response (0.21% of trials) were excluded. One participant was an outlier in terms of biasing (error rate difference between low- and high-value trials fell more than two interquartile ranges above the upper quartile) and was excluded from further analyses.

## Electrophysiological data analysis

Continuous EEG data from 128 scalp electrodes were acquired using an ActiveTwo system (BioSemi, The Netherlands) and digitized at 1024 Hz. Offline analyses were performed using in-house MATLAB scripts (MathWorks, Natick, MA) using data reading, channel interpolation, and topographic plot functions from the EEGLAB toolbox (*Delorme and Makeig, 2004*). EEG data were low-pass filtered by convolution with a 137-tap Hanning-windowed sinc function designed to provide a 3 dB corner frequency of 37 Hz with strong attenuation at the mains frequency (50 Hz), and detrended. The data were epoched from −150 to 2450 ms relative to the cue onset. We identified and interpolated (spherical splines) channels with excessively high variance with respect to neighboring channels and channels that saturated or flat-lined during a given block. The data were then average-referenced, and trials were rejected upon detection of artifacts between cue and response (on any channel with magnitude >70 µV, or 50 µv for the selected motor channels used in our analyses). Then, to mitigate the effects of volume conduction across the scalp, current source density transformation was applied to the single-trial epochs (*Kayser and Tenke, 2006*; *Kelly and O'Connell, 2013*). Shorter cue-locked (−150 to 1500 ms), stimulus-locked (−1000 to 650 ms), and response-locked (−400 to 210 ms) ERPs were then extracted from the longer epochs, average-referenced and baseline corrected to the 100-ms window following the cue. The cue- and stimulus-locked LRP was calculated as the difference in ERP between electrodes at standard 10–20 sites C3 and C4 (*Gratton et al., 1988*), by subtracting the ERP ipsilateral to the correct response from the contralateral ERP.

Beta-band activity was measured using a short-time Fourier transform applied to 300-ms windows stepped by 25 ms at a time, and by taking the mean amplitude in the range 14–30 Hz. We restricted our measurements to the beta band as opposed to including both mu and beta (*Kelly et al., 2021*) to avoid any potential interference from posterior alpha-band activity which is known to lateralize in situations where attention can be guided to the left or right. We found posterior lateralization to be minimal in the beta-band amplitude, and while there was an appreciable slope difference, this was clearly separated from the motor-related areas (see *Figure 5A*). To ensure precise measurements for model constraints, beta was measured from electrodes selected per individual based on exhibiting the strongest decrease at response relative to cue or stimulus onset. Standard sites C3/C4 were selected by default where difference topography foci were close and symmetric (9 of 17 participants), and otherwise electrodes were selected among those proximal to the foci based on their exhibiting smooth decline in their amplitude timecourses from cue to response. Where uncertain, preference was given to symmetry across hemispheres and electrodes that reached a common threshold across conditions at response.

For these individually selected electrodes (marked in *Figure 5A*), the contralateral beta just prior to response (−50 ms) reached a threshold across conditions (*Figure 5B*; the error bars in *Figure 2B* break this down further into value and response conditions). The ipsilateral beta diverged between the blocked high coherence and the other conditions, indicating a closer race for the most speed-pressured condition. When the standard C3/C4 sites were instead selected, however, we found an offset between the blocked conditions and the interleaved conditions (*Figure 5C*). This was unexpected, but not entirely surprising due to the fact that the blocked and interleaved sessions were performed on different days for all participants, and the different demands potentially resulted in some global changes in measured beta amplitude not directly related to motor preparation. The inset topographies show the overall difference in beta amplitude between the two sessions at response; the difference does not appear to be of motor origin. As this difference was evident to a similar degree before the stimulus onset, we recalculated the beta starting points and slopes with the C3/C4 electrodes after first subtracting the offset between the two sessions at −50 ms from response from all beta traces. We found that the calculated neural constraints were similar regardless of electrode choice (*Table 5*). The starting levels were almost identical except for a small difference in the low-coherence-blocked levels both contralateral and ipsilateral to high value. The steeper ipsilateral slope was also maintained and the difference relative to contralateral slope had a similar magnitude. Due to our desire to obtain the clearest view of motor activity possible, we used the individually selected electrodes in our modeling and analyses.

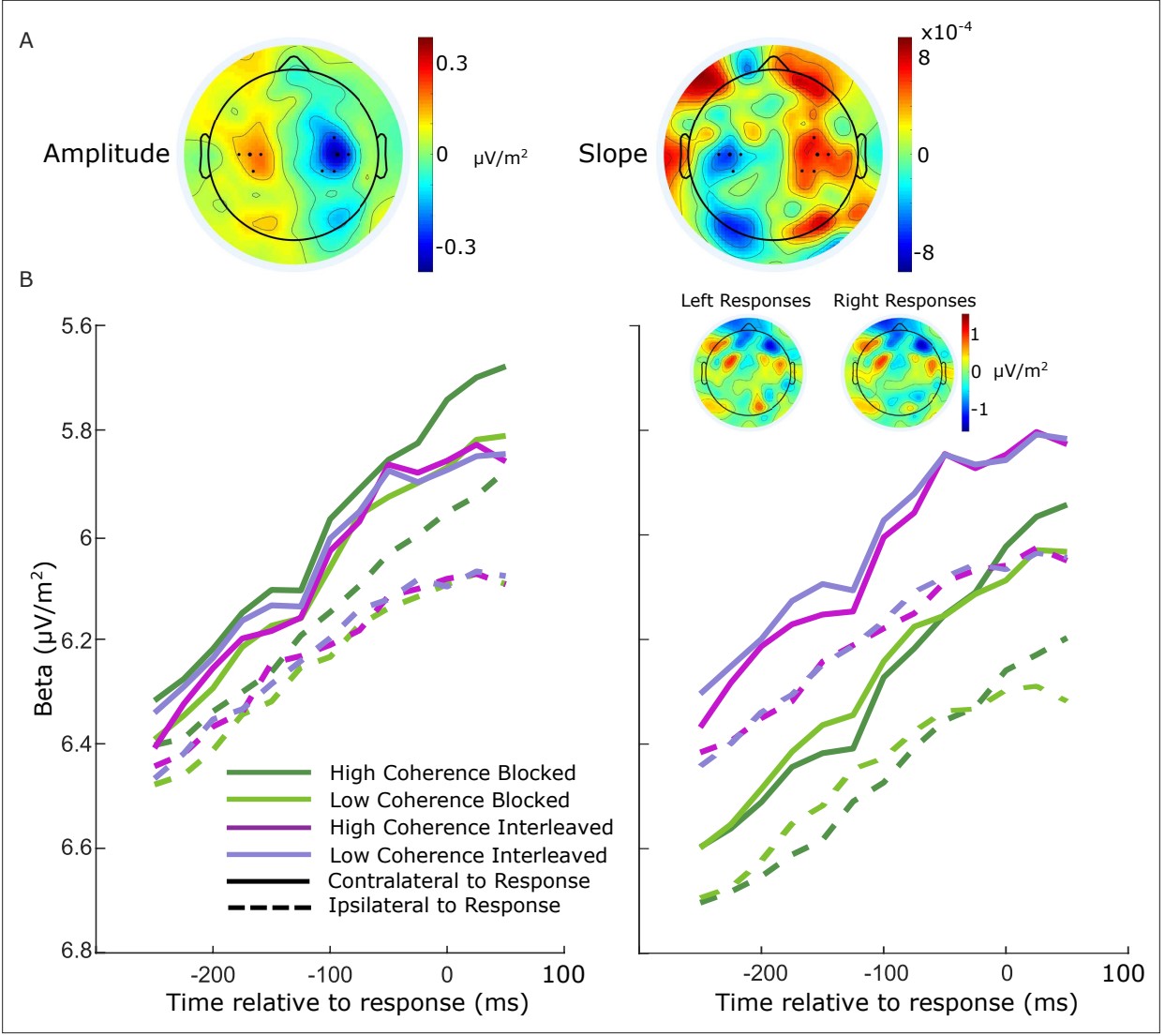

**Figure 5.** Electrode selection for beta analysis. (**A**) Topographies of the difference between left- and right-cued trials for beta amplitude at 750 ms relative to amplitude at the cue, and slope from 700 to 800 ms after the cue. Standard sites C3/C4 are marked with large black dots, while other electrodes that were selected for certain individuals are marked with smaller dots. (**B**) Response-locked beta contralateral (solid) and ipsilateral (dashed) to response for the four conditions with individually selected electrodes. (**C**) Same as B, but with standard sites C3/C4 selected for all participants. Topographies show the average difference in beta amplitude between blocked and interleaved conditions at −50 ms relative to response, for right and left responses separately.

**Table 5.** Scaled beta start points and slopes for individually selected electrodes and C3/C4.

| Parameter | Individually selected | | | C3/C4 | | |
|---|---|---|---|---|---|---|
| | High coherence | Low coherence | Interleaved | High coherence | Low coherence | Interleaved |
| $Z_c$ | 0.33 | 0.3 | 0.2 | 0.33 | 0.35 | 0.2 |
| $Z_i$ | 0.14 | 0.003 | 0 | 0.12 | 0.06 | 0 |
| $U_c$ | 1.33 | 1.06 | 1.26 | 1.24 | 0.95 | 1.17 |
| $U_i$ | 1.78 | 1.66 | 1.76 | 1.66 | 1.61 | 1.63 |

## Statistical approach

Repeated measures analyses of variance (ANOVAs) with both Value and Regime/Conditions included as appropriate were used to test for differences in behavioral and neural amplitude and slope measures, and followed up with pairwise, false discovery rate-corrected *t*-tests using the Python package pingouin (*Vallat, 2018*). Given the study's focus on mechanisms common to the various conditions, we state main effects of value in the main text, and address regime effects in the figure legends. The onsets for the beta signals were calculated using a jackknife procedure in which the traces were computed for the average signals of 16 participants at a time, with each participant systematically excluded in turn, to compute the first time at which it exceeded 20% of the response threshold for that subgroup. The standard errors of each condition were then scaled up by 16 and a repeated measures ANOVA was conducted.

## Modeling

We fit several alternative bounded accumulation models to the data for comparison. In the neurally constrained models, to linearly scale the pre-stimulus beta signals we defined the lowest 'starting level' and bound to be 0 and 1, respectively. The mean contralateral beta amplitude 50 ms before response was mapped to the bound, while the condition with the lowest beta amplitude 750 ms after the cue was mapped to 0.

In the standard DDM (*Ratcliff and McKoon, 2008*), noisy evidence accumulated in one dimension as in *Equation 7* but without drift-rate bias; $B = 0$. Evidence accumulated to a fixed bound ($a$) which varied across the three regimes, and there was no urgency. In each regime, we allowed a separate biased mean starting point of accumulation ($z_b$), with uniformly distributed variability $s_z$ so that:

$$x(0) = \pm z_b + U(-s_z/2, s_z/2) \tag{16}$$

where $z_b$ has a positive sign for high-value trials, and a negative sign for low-value trials. A nondecision time ($T_{er}$) parameter (different for each regime) was also perturbed by uniformly distributed variability ($s_t$) and added to the decision time to obtain the final RT. There were two drift-rate parameters—one for each coherence—that were constant over time and common across the regimes, but varied from trial to trial in a Gaussian distribution with standard deviation $\eta$. By convention, the square root of the diffusion coefficient, or standard deviation of the within-trial noise, $\sigma$, was fixed at 0.1 and acted as a scaling parameter for the model.

We fit each model to 16 RT distributions (*Figure 4D, E*): correct and error responses for high- and low-value trials across the four conditions. We partitioned each distribution into six bins bounded by the 0.1, 0.3, 0.5, 0.7, and 0.9 quantiles. Models were fit by minimising the chi-squared statistic $G^2$, between the real quantiles and those obtained from Monte-Carlo simulated RT distributions:

$$G^2 = 2\left(\sum_{c=1}^{4}\sum_{v=1}^{2} N_{c,v}\left[\sum_{o=1}^{2}\sum_{q=1}^{6} p_{c,v,o,q} log \frac{p_{c,v,o,q}}{\pi_{c,v,o,q}}\right]\right) \tag{17}$$

where $p_{c,v,o,q}$ and $\pi_{c,v,o,q}$ are the observed and predicted proportions of responses in bin $q$, bounded by the quantiles, of outcome $o$ (correct/error) of condition $c$ (coherence × Blocked/Interleaved) and value $v$ (high/low), respectively. $N_{c,v}$ is the number of valid trials per condition and value.

In the model simulations the urgency signals were defined to equal their scaled (750 ms post-cue) beta levels at 100 ms prior to stimulus onset time. In the experiment, stimulus onset corresponded to 850 or 900 ms post-cue; thus, we started the stimulus-evoked accumulation with a 50-ms delay on half of the trials and adjusted the RTs accordingly. For the IE models, the shape function $\vartheta(t)$ was obtained in our simulations by numerical integration. We searched the parameter space using the particle swarm optimization algorithm (*Kennedy and Eberhart, 1995*) as implemented in MATLAB, initialized with a number of swarms equal to 10 times the number of parameters to be estimated. To aid convergence we set the same random seed for each simulation within a search, which comprised 20,000 trials per value per condition. Because there was randomness associated with the optimization we ran it at least four times for each model. We then obtained a final $G^2$ for each parameter vector by running a simulation with 2,000,000 trials and initialized with a different seed, and selected that with the lowest value. We performed model comparison using AIC, which penalises models for complexity:

$$AIC = G^2 + 2k \tag{18}$$

where $k$ is the number of free parameters. We also calculated the Akaike weights (*Burnham and Anderson, 2002*; *Wagenmakers and Farrell, 2004*), which can be cautiously interpreted as providing a probability that model $i$ is the best model in the set:

$$W_i\left(AIC\right) = \frac{e^{\frac{-1}{2}\Delta_i\left(AIC\right)}}{\sum_{i=1}^{k} e^{\frac{-1}{2}\Delta_i\left(AIC\right)}} \tag{19}$$

where $\Delta_i\left(AIC\right)$ is the difference in AIC between model $i$ and the best-fitting model.

The simulated DVs for comparison with the real LRP were obtained by subtracting the average DV of the incorrect option from the correct option, time-locked to stimulus onset. We did not make the simulations fall back to zero upon bound crossing, and so the signals continue to build and become less comparable to the real average LRP once it peaks and falls due to responses being made. Initially, we had allowed the possible range of burst durations to be a free parameter in the BurstIE model and obtained several equally good fits in which this parameter was spread over a wide range of values, trading off with the bias magnitude. We thus decided to constrain this parameter to correspond to the real LRP as closely as possible, with the understanding that within our framework we could not be certain of its exact form. We fit the model four times with the burst duration range set to 30, 50, 70, and 90 ms, and compared the time between burst onset and the low-value turnaround in the real LRP (53.7 ms) to those in the simulations. Finding the 70-ms duration range gave the closest match (52 ms), we then adjusted the duration range parameter holding all others constant to obtain a 54-ms simulated LRP duration when the range parameter was set to 72 ms. We adopted this value in all further fits to the BurstIE and BurstSE models.

## Acknowledgements

The authors thank Louisa Spence for data collection. This study was funded by the European Union's Horizon 2020 research and innovation programme under the Marie Skłodowska-Curie grant agreement No. 842143, the European Research Council Starting Grant No. 63829, the European Research Council Consolidator Grant IndDecision—865474, the Irish Research Council (GOIPD/2017/1261), and by Science Foundation Ireland Grant No. 15/CDA/3591. Most of the model fitting was performed on the Lonsdale cluster which is funded through grants from Science Foundation Ireland and maintained by the Trinity Centre for High Performance Computing (Research IT, Trinity College Dublin).

## Additional information

### Competing interests

Redmond G O'Connell: Reviewing editor, *eLife*. The other authors declare that no competing interests exist.

### Funding

| Funder | Grant reference number | Author |
| --- | --- | --- |
| H2020 Marie Skłodowska-Curie Actions | 842143 | Elaine A Corbett |
| Science Foundation Ireland | 15/CDA/3591 | Simon P Kelly |
| H2020 European Research Council | Starting Grant No 63829 | Redmond G O'Connell |
| H2020 European Research Council | Consolidator Grant IndDecision, No 865474 | Redmond G O'Connell |
| Irish Research Council | GOIPD/2017/1261 | Elaine A Corbett |

The funders had no role in study design, data collection, and interpretation, or the decision to submit the work for publication.

## Author contributions

Elaine A Corbett, Conceptualization, Software, Formal analysis, Funding acquisition, Investigation, Methodology, Writing – original draft, Project administration, Computational model development and fitting; L Alexandra Martinez-Rodriguez, Conceptualization, Investigation, Project administration; Cian Judd, Investigation, Methodology; Redmond G O'Connell, Conceptualization, Resources, Funding acquisition, Methodology, Writing – review and editing; Simon P Kelly, Conceptualization, Resources, Software, Funding acquisition, Methodology, Writing – review and editing, Development of computational models

## Author ORCIDs

Elaine A Corbett ⓘ http://orcid.org/0000-0001-7009-5867
L Alexandra Martinez-Rodriguez ⓘ http://orcid.org/0000-0002-9739-0572
Redmond G O'Connell ⓘ http://orcid.org/0000-0001-6949-2793
Simon P Kelly ⓘ http://orcid.org/0000-0001-9983-3595

## Ethics

Informed consent and consent to publish were obtained. Procedures were approved by the Ethics Committee of the School of Psychology at Trinity College Dublin, and the Human Research Ethics Committee for the Sciences, at University College Dublin (LS-17-88).

## Decision letter and Author response

Decision letter https://doi.org/10.7554/eLife.67711.sa1
Author response https://doi.org/10.7554/eLife.67711.sa2

---

# Additional files

## Supplementary files

• Transparent reporting form

## Data availability

The data and code have been published on Figshare plus at the following doi: https://doi.org/10.25452/figshare.plus.c.6338693.v1.

The following dataset was generated:

| Author(s) | Year | Dataset title | Dataset URL | Database and Identifier |
|---|---|---|---|---|
| Corbett EA, Martinez-Rodriguez LA, Judd C, O'Connell RG, Kelly SP | 2022 | Multiphasic Value Biases in Fast-Paced Decisions | https://doi.org/10.25452/figshare.plus.c.6338693.v1 | Figshare, 10.25452/figshare.plus.c.6338693.v1 |

---

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
