## [Editor Report]

This study reports important insights into perceptual decisions made under high temporal pressure, by revealing a complex interplay of different biasing mechanisms unfolding during decision formation. The neurally-constrained approach adopted in the study, using the dynamics of EEG correlates of decision-making, provides convincing evidence in favor of the proposed biasing mechanisms. This generic approach could be applied in future work in other contexts, using other types of neural data, to refine, validate or falsify candidate models of human and animal behavior.

---

## [Decision Letter]

**Decision letter after peer review:**

Thank you for submitting your article "Multiphasic value biases in fast-paced decisions" for consideration by *eLife*. Your article has been reviewed by 2 peer reviewers, and the evaluation has been overseen by Valentin Wyart as the Reviewing Editor and Michael Frank as the Senior Editor. The following individuals involved in review of your submission have agreed to reveal their identity: Anna-Lena Schubert (Reviewer #1).

The reviewers have discussed their reviews with one another, and the Reviewing Editor has drafted this to help you prepare a revised submission. Overall, we have found your brain-behavior approach to model identification to be innovative. The fact that it allows for the comparison of more complex models that include parameters which could not be identified reliably from behavior alone is important and may stimulate new work. Furthermore, your finding that value biases in value-based decisions are best explained by a complex, "multiphasic" pattern of effects rather than a single mechanism (as often postulated by standard drift-diffusion accounts) is important for the field. Despite these merits, we have identified a number of essential revisions that should be made to provide stronger support for the reported findings. These revisions do not require additional data, but they will require additional analyses of the data. The separate reviews are also provided for more detailed information about the required revisions, but they do not require a point-by-point response.

Essential revisions:

1) Some of the main findings need further support, and alternative interpretations of the data need to be considered explicitly in the manuscript. First, because you present the multiphasic account against a standard account, it is very important to test your multiphasic model against the standard drift-diffusion model where value biases are reflected in shifts of the starting point. The aim of this analysis is to quantify how much better your proposed model performs in comparison to such a standard model. This would be an important point to make to argue for the benefits of your neurally-constrained modeling approach. Second, the claim that cue-evoked motor β reflects urgency deserves additional support. It is currently more taken as a given than actually demonstrated. This claim should be further corroborated using aspects of the data (e.g., by showing that response deadlines modulate this signal). Related to this question, it is currently unclear whether the pre-stimulus pattern seen in motor β feeds into the decision making process itself. Indeed, it is not hard to imagine why left and right pointing arrows directly trigger motor activity (i.e., priming), but it does not necessarily imply that such activity leaks into the decision process that may occur upstream from response preparation.

2) The manuscript is currently very dense, resulting in several statements that currently look unjustified and that should be expanded and supported by the existing literature. First, the introduction of urgency signals is extremely short in the current version of the manuscript. You should elaborate and clarify the idea when it is first introduced. Second, we did not understand the claim that classical evidence accumulation models are primarily developed to study slow, deliberative perceptual decisions. Much of this work is based on the categorization of random-dot motion stimuli, which I would not call a slow, deliberative decision. You should revise this statement or clarify your view in the revised manuscript. Third, one of the main motivations of the study is that we currently lack a model that can explain fast-paced value-based decisions, mostly because the constant drift-rate assumption in evidence accumulation models seems invalid. This conjecture is actually very similar to the existing literature on response conflict, where performance in conflict tasks (such as Stroop, Flanker, etc.) is best modelled using a time-varying drift rate. Because this literature is not introduced nor discussed in the current version of the manuscript, we wonder to what extent the reported findings may reflect the same process, i.e., that the value cue "primes" a response, which then has to be suppressed in favor of the correct response. It would be very important to consider explicitly in the revised manuscript the possibility that the motor β effect may reflect priming rather than urgency. There are likely differences between the two accounts, but they should be discussed explicitly. Fourth, given that some of the authors previously made a strong case that evidence accumulation is reflected in the centro-parietal positivity (CPP), it is a bit surprising not to see any mention of this notion in the current manuscript, and instead only read about motor β activity and lateralized readiness potentials. There are likely reasons for the lack of analysis of the CPP, but this would deserve some justification in the revised manuscript.

3) While the general modeling framework is innovative and highly appealing, there are modeling issues with the analyses reported in the manuscript. A first potential limit of the findings is the sample size and composition. With only 18 participants (one of which was a co-author of this manuscript, see point 4 below), the robustness of the authors' modeling results remains an open question. This low sample size is particularly problematic because the modeling approach used by the authors currently ignores between-participant variability and relies on a fixed-effects approach. It is unclear why this was done (rather than fitting data separately per participant or fitting the data using a hierarchical model), because it induces substantial variance in the fits caused by between-participant differences. This should be corrected in the revised manuscript. Furthermore, fitting the model separately for each participant (possibly using a hierarchical approach) would allow testing if the best-fitting model identified in Table 2 provides the best account of the data for most participants. In practice, we suggest fitting the model separately for each participant and to report for how many participants each of the implemented models provided the best account of the data. Moreover, we would suggest reporting Akaike weights (or relative likelihoods) as described in Wagenmakers & Farrell, 2004.

Wagenmakers, E.-J., & Farrell, S. (2004). AIC model selection using Akaike weights. Psychonomic Bulletin & Review, 11(1), 192-196. https://doi.org/10.3758/BF03206482

4) Data curation: it would be safer to exclude the data from the one participant who was also a co-author of this study. Similarly, it would be safer to to not include the data from the second accidentally recorded experimental session in the data to rule out training effects.

5) The choice of the experimental design could be better motivated in the introduction. You write in the introduction that you "primarily focused on the value biasing dynamics in common across these challenging regimes". So it feels like task conditions with distinct value differences could have been more instructive than the proposed experimental design. A design with distinct value differences would also distinguish between your favored account (where different levels of value should parametrically affect motor β) and a simple priming account (where value should not matter). The revised manuscript should better explain why the chosen experimental conditions are important for revealing the main findings.

6) A Bayes Factor (BF) of 0.87 does not afford the conclusion that there was no significant difference between low-coherence conditions. Instead, it indicates that the data do not contain enough evidence to draw any conclusions. The manuscript should be corrected accordingly.

*Reviewer #1 (Recommendations for the authors):*

There are a few issues that the authors need to address before I can recommend publication of the manuscript:

1. The introduction of urgency signals is extremely short, probably due to word limits. Nevertheless, I would suggest to elaborate or at least clarify the idea when it is first introduced (l. 45-46).

2. There may be a typo in the legend of Figure 1 (check deadlines).

3. I do not agree that a BF of 0.87 allows the conclusion that there was no significant difference between low-coherence conditions (p. 3). Instead, it indicates that the data do not contain enough evidence to draw any conclusions.

4. The authors should explain how the time window used to determine the β threshold was chosen (l. 232-233).

5. I really think that the low sample size is a limitation of the study that needs to be addressed. I suggest to fit the model separately for each participant and to report for how many participants each of the implemented models provided the best account of the data. Moreover, I would suggest to report Akaike weights (or relative likelihoods) in addition to AICs (Wagenmakers & Farrell, 2004).

6. I would urge the authors to exclude the data from the one participant who was also a co-author of this study.

7. The did not understand the color differentiation in Figure 4, but presume that lightly shaded red indicates high value, low coherences, whereas saturated red indicates high value, high coherence. If this were the case, the complex figure may become more easily comprehensible if these line types were integrated into a single color legend.

8. Experimental errors happen. Nevertheless, I suggest to not include the data from the second accidentally recorded experimental session in the data to rule out training effects (l. 453-454).

9. I think that it would be informative to test the authors' model against a standard diffusion model where decision biases were reflected in shifts of the starting point. How much better would the authors' model perform in comparison to such a standard model? I believe that this might be an important point to make to argue for the benefits (with regard to the efforts) of the authors' modeling approach.

Wagenmakers, E.-J., & Farrell, S. (2004). AIC model selection using Akaike weights. Psychonomic Bulletin & Review, 11(1), 192-196. https://doi.org/10.3758/BF03206482

*Reviewer #2 (Recommendations for the authors):*

– I felt the entire manuscript was a bit dense. For example, evidence accumulation models are described only very briefly in the introduction, making it hard to grasp exactly how the current model differs from classical models (such as e.g. the drift diffusion model). I think this limits the extent to which a broad readership will appreciate the current paper.

– Could the authors explain how the model dissociates between B(t) and u1 (or u2) because (as I understand it) both reflect a constant addition to the signal. Is this only achieved because B is modelled as a sudden burst, or is that not necessary? Related to that, did the authors conduct parameter recovery?

– Given that the authors previously made a strong case that evidence accumulation is reflected in the CPP, I was a bit surprised not to see any mention to that and instead only read about motor β and LRPs. I can imagine why, but this deserves some explanation.

– Line 30-32: I did not understand the claim that classical evidence accumulation models are primarily developed to study slow, deliberative perceptual decisions. Much of the EAM work is based on random dot motion work, which I would not call a slow, deliberative decision.

– Line 93: how can you press a mouse with both thumbs?

– The figures could still be improved a bit. For example, 2D doesn't have a y-axis label; the lower topo in 2A misses a slight piece of its label; Figure 3 would be easier to follow with a bit more panel labels; it took me some time before I figured out the legend in Figure 4 (especially correct vs error)

– Could the authors confirm (and if so explicitly state) that the boundary in their model was fixed to an arbitrary value?

– Line 376: The term "negative drift rate bias in urgency" falls a bit out of the blue. This could be introduced better, and/or perhaps explicitly called as such around line 120.

– In line with *eLife*'s policies: is the data and code freely accessible?

[Editors' note: further revisions were suggested prior to acceptance, as described below.]

Thank you for resubmitting your work entitled "Multiphasic value biases in fast-paced decisions" for further consideration by *eLife*. Your revised article has been evaluated by Michael Frank (Senior Editor) and Valentin Wyart (Reviewing Editor).

You have provided detailed responses to the essential revisions outlined in our former decision letter. We had now time to consider your responses, we are sorry for the delay and thank you for your patience.

We found that you provided convincing and in-depth answers and changes to the manuscript. In particular, you now test your model against the classical drift-diffusion model (DDM) found in the literature. The fact that the dynamics of β amplitude vary as a function of speed pressure makes it unlikely that it reflects automatic priming. The paper also reads better now that you have added several additional explanatory paragraphs where needed, and you provide a more accurate description of the existing literature.

Two responses in your point-by-point rebuttal letter required more thought: 1. your decision to stick to model fits in terms of group-level averages rather than individual participants, and 2. your decision not to exclude the participant who was not naive to the purpose of the study.

Regarding point 1, we found that you indeed provided convincing explanations as to why you did not follow the reviewers' suggestion to perform individual fits to the data. Nevertheless, as a minor revision request, I would find it more appropriate that you introduce the fact that you will be fitting group-level averages rather than individual participants at the beginning of the "Model fits" section (starting at line 316). You should describe in a few sentences the rationale (which you outline clearly in the rebuttal letter) as to why you are using such a fitting approach over the more standard (individual participant) approach. You make totally valid points in the rebuttal letter and we think they should be spelled out at the onset of the "Model fits" section. These are important points for readers. It is good that you now provide the "jackknifing procedure for model comparison" subsection at the end of the Results section which, we agree with you, provides additional and supporting evidence that your results are robust and not driven by an individual subject.

Regarding point 2, we tend to disagree that your study of biasing effects does not create issues when testing participants who are not blind to the purpose of the study. Although the biases you describe may not be explicit and fully voluntary, being informed about the purpose of the study may (even if unlikely) create effects that are not present in naive subjects. Because reviewers specifically asked for you to remove the participant in question, we took some time to consider your decision to keep this participant – in light also of your response to point 1 above. With the additional data that you report in the revised manuscript, and in the rebuttal letter, we agree that it is highly unlikely that a single participant drives the group-level effects that are reported in the study. Therefore, we agree that this non-naive participant can be kept in the dataset.

---

## [Author Response]

Essential revisions:1) Some of the main findings need further support, and alternative interpretations of the data need to be considered explicitly in the manuscript. First, because you present the multiphasic account against a standard account, it is very important to test your multiphasic model against the standard drift-diffusion model where value biases are reflected in shifts of the starting point. The aim of this analysis is to quantify how much better your proposed model performs in comparison to such a standard model. This would be an important point to make to argue for the benefits of your neurally-constrained modeling approach.

We have now added the standard diffusion decision model (DDM) with starting point shifts to the comparison in the revised paper (see results in Table 2 and Figure 4—figure supplement 4, presented in the text from line 336; methods from line 617). The DDM performs very poorly relative to all of the neurally-constrained alternatives. Indeed, this poor performance was expected given there are substantive qualitative features of the data that the DDM is not equipped to capture such as fast biased errors, as has been established before (Diederich & Busemeyer 2006, Afacan-Seref et al., 2018, Kelly et al., 2021).

Second, the claim that cue-evoked motor β reflects urgency deserves additional support. It is currently more taken as a given than actually demonstrated. This claim should be further corroborated using aspects of the data (e.g., by showing that response deadlines modulate this signal). Related to this question, it is currently unclear whether the pre-stimulus pattern seen in motor β feeds into the decision making process itself. Indeed, it is not hard to imagine why left and right pointing arrows directly trigger motor activity (i.e., priming), but it does not necessarily imply that such activity leaks into the decision process that may occur upstream from response preparation.

We thank the reviewers for this point; we had not provided a complete account of all of the evidence indicating that β signals reflect contributions from urgency that feed into the decision variables. As both the left and right arrow cues were presented simultaneously on every trial with isoluminant color we would not expect a bottom-up lateralization in response to their physical appearance as might occur following the onset of a single arrow cue, and indeed, we found similar patterns of biased urgency for probability cues implemented by a dot-color change rather than arrows (Kelly et al., 2021). As we now more clearly lay out, we have seen in our previous work that these anticipatory motor signals in the mu and β bands begin their dynamic buildup much later in the absence of speed pressure (see Author response image 1) and from a lower starting point, and under speed pressure reach approximately halfway to the threshold in agreement with a halving of the decision bounds estimated by behavioral models (Kelly et al., 2021). In contrast with that prior work, here the stimuli and deadlines were designed so that the participants would feel a similar level of speed pressure with the longer deadline combined with lower discriminability, as with the shorter deadline and higher discriminability. It was therefore designed not to find large differences in these signals between the three regimes. Nevertheless, and quite remarkably, we find that when we shuffle the assignment of particular β measurements to different regimes the model fit becomes substantially poorer (Table 2), strongly suggesting that the specific levels of advance preparation reflected in β do feed into the decision process itself.

**Author response image 1. sa2fig1:** Adapted from Kelly, Corbett & O’Connell 2021. Mu/Beta motor preparation signals between a cue indicating the more probable alternative and evidence onset. The Deadline condition (red) had more speed pressure than the other two. Dynamic buildup of motor preparation began earlier in the deadline condition. The markers indicate the points at which the buildup rate of the beta signal in each regime significantly rises above zero.

Below we quote the updated section of the results which more carefully describes these prior findings and our reasoning in interpreting these signals as we do (line 122). We further consider the idea of priming in our Discussion, as noted in response to point 2 below.

“EEG Signatures of Motor Preparation. Decreases in spectral amplitude in the β band (integrated over 14-30Hz) over motor cortex reliably occur with the preparation and execution of movement (Pfurtscheller, 1981). When the alternative responses in a decision task correspond to movements of the left and right hands, the signal located contralateral to each hand leading up to the response appears to reflect effector-selective motor preparation that is predictive of choice (Donner et al., 2009). Furthermore, before the onset of sensory evidence the ‘starting levels’ of the signals reflect biased motor preparation when prior expectations are biased (de Lange et al., 2013), and are higher under speed pressure for both alternatives (Kelly et al., 2021; Murphy et al., 2016; Steinemann et al., 2018), implementing the well-established decision variable (DV) adjustments assumed in models (Bogacz et al., 2010; Hanks et al., 2014; Mulder et al., 2012). The signal contralateral to the chosen hand then reaches a highly similar level at response irrespective of stimulus conditions or response time, consistent with a fixed, action-triggering threshold (Devine et al., 2019; Feuerriegel et al., 2021; Kelly et al., 2021; O’Connell et al., 2012; Steinemann et al., 2018). The level of β before stimulus onset also predicts response time and its poststimulus buildup rate scales with evidence strength, underlining that this signal reflects both evidence-independent and evidence-dependent contributions to the decision process (Steinemann et al., 2018). Thus, we can interpret the left- and right-hemisphere β as reflecting two race-to-threshold motor-preparation signals whose buildup trace the evolution of the decision process from stimulus anticipation through to the response (Devine et al., 2019; Kelly et al., 2021; O’Connell et al., 2012).

Here, prior to stimulus onset, motor preparation (decrease in β amplitude) began to build in response to the value cue, first for the high-value alternative and later for the low-value alternative (F(1,16)=15.8, p=.001, partial η^2^=0.5 for jackknifed onsets, Figure 2A), and continued to build for both alternatives after stimulus onset. Consistent with prior work suggesting an action-triggering threshold, the signal contralateral to the chosen hand reached a highly similar level at response irrespective of cue-type, coherence or regime (Figure 2B). Before the stimulus onset, rather than generating a stable starting level bias the motor preparation signals continued to increase dynamically. This replicates similar anticipatory buildup observed in a previous experiment with prior probability cues, and does not reflect an automatic priming due to the cue because its dynamics vary strategically with task demands such as speed pressure (Kelly et al., 2021). Thus, we take the anticipatory buildup to reflect dynamic urgency that, independent of but in addition to the evidence, drives the signals towards the threshold (Churchland et al., 2008; Hanks et al., 2014; Murphy et al., 2016; Steinemann et al., 2018; Thura and Cisek, 2014).”

2) The manuscript is currently very dense, resulting in several statements that currently look unjustified and that should be expanded and supported by the existing literature. First, the introduction of urgency signals is extremely short in the current version of the manuscript. You should elaborate and clarify the idea when it is first introduced.

In addition to the new treatment of the urgency signals added above, we have added a more elaborated introduction to the existing models, including models of urgency, at the beginning of our model development section (line 185):

“Model Development. We next sought to construct a decision process model that can capture both behavior and the motor preparation dynamics described above. Probably the most widely-used evidence-accumulation model for two-alternative decision making is the diffusion decision model (DDM, Ratcliff, 1978), which describes a one-dimensional stationary evidence accumulation process beginning somewhere between two decision bounds and ending when one of the bounds is crossed, triggering the associated response action. The time this process takes is known as the decision time, which is added to a nondecision time (accounting for sensory encoding and motor execution times) to produce the final RT. This model has been successfully fit to the quantiles of RT distributions (e.g. Figure 1—figure supplement 1) for correct and error responses across a wide range of perceptual decision contexts. Traditionally, value biases can be incorporated into this framework by either biasing the starting point closer to one bound than the other or biasing the rate of evidence accumulation, the former of which generally better describes behavior (Ratcliff and McKoon, 2008). However, researchers have found that when there is a need to respond quickly a stationary evidence accumulation model is not sufficient to capture the pattern of value biases in behavior, which exhibits a dynamic transition from early, valuedriven responses to later evidence-based ones. Accounting for this fast value-biased behavior in a DDM framework has instead required a non-stationary drift rate; either a dual phase model with an initial value-based drift rate transitioning to a later evidence-based one (Diederich and Busemeyer, 2006), or combining a constant drift rate bias with a gradually increasing sensory evidence function (Afacan-Seref et al., 2018). Alternatively, Noorbaloochi et al. (2015) proposed a linear ballistic accumulator model with a probabilistic fast guess component that was driven by the value information. However, in each of these approaches evidence accumulation begins from a stable starting point, meaning they could not account for the dynamic biased anticipatory motor preparation activity.”

“Combined urgency + evidence-accumulation model: As noted above, we interpreted the anticipatory β changes to be reflective of a dynamic urgency driving the motor preparation for each alternative towards its threshold, independent of sensory evidence. Urgency has been found to be necessary to explain the more symmetrical RT distributions found in many speed-pressured tasks as well as the sometimes strong decline in accuracy for longer RTs in these conditions. Urgency has been implemented computationally in a variety of ways, reviewed in detail by Smith & Ratcliff (2021) and Trueblood et al. (2021). While models assuming little or no accumulation over time characterize urgency as a “gain” function that multiplies the momentary evidence, models centered on evidence accumulation assume that urgency adds to cumulative evidence in a decision variable with a fixed threshold, which is mathematically equivalent to a bound on cumulative evidence that collapses over time (Drugowitsch et al., 2012; Evans et al., 2020; Hawkins et al., 2015; Malhotra et al., 2017). The latter, additive urgency implementation is consistent with neurophysiological signatures of urgency found across multiple evidence strengths including zero-mean evidence (Churchland et al., 2008; Hanks et al., 2011) and provides the most natural interpretation of the β signals here due to their anticipatory, pre-stimulus buildup before evidence accumulation was possible. We therefore drew on a recently proposed model for decisions biased by prior expectations with two discrete levels: the onedimensional accumulation of stimulus-evoked activity (noisy sensory evidence and bias) is fed to a ‘motor’ level where it is combined additively with evidence-independent buildup components that linearly increase with time (Murphy et al., 2016; Steinemann et al., 2018) to generate the motor-level DVs engaging in a race to the bound (Kelly et al., 2021, Figure 3A).”

Second, we did not understand the claim that classical evidence accumulation models are primarily developed to study slow, deliberative perceptual decisions. Much of this work is based on the categorization of random-dot motion stimuli, which I would not call a slow, deliberative decision. You should revise this statement or clarify your view in the revised manuscript.

Thank you for spotting this – It is true that decisions about stimuli like the dots are made within a second, but in the domain of perceptual decisions, our task is considerably faster than the typical task settings used which have had less speed pressure. We have clarified and revised the text to read

‘…through the study of perceptual decisions with low to moderate speed pressure’

Third, one of the main motivations of the study is that we currently lack a model that can explain fast-paced value-based decisions, mostly because the constant drift-rate assumption in evidence accumulation models seems invalid. This conjecture is actually very similar to the existing literature on response conflict, where performance in conflict tasks (such as Stroop, Flanker, etc.) is best modelled using a time-varying drift rate. Because this literature is not introduced nor discussed in the current version of the manuscript, we wonder to what extent the reported findings may reflect the same process, i.e., that the value cue "primes" a response, which then has to be suppressed in favor of the correct response. It would be very important to consider explicitly in the revised manuscript the possibility that the motor β effect may reflect priming rather than urgency. There are likely differences between the two accounts, but they should be discussed explicitly.

As noted in response to point 1, we now have included further explanation as to why we think the motor β effect represents urgency as opposed to priming. Specifically, in previous work we saw a much later dynamic buildup in conditions with reduced speed pressure (see Author response image 1), and the simultaneous presentation of the left and right arrow cues with isoluminant color would not be expected to cause this kind of bottom-up lateralization. Nonetheless, we thank the reviewer for highlighting this interesting parallel with conflict tasks, and indeed, a previous study from our lab also pointed out this parallel (Afacan-Seref et al. 2018). In particular, the deflection towards the high value alternative seen in the LRP appears similar to that seen in conflict tasks, although in the current case it reflects a topdown influence as there is no physical difference in the appearance of the stimulus, except for the symmetric arrow cues which first appeared >800ms prior. We have added the following paragraph to the Discussion:

“The dynamic shift from value to evidence-driven accumulation is reminiscent of conflict tasks, for which a stationary drift rate is similarly insufficient to describe the observed behavioral patterns. In these tasks, the context in which a perceptual stimulus is presented (i.e. features of the stimulus that are irrelevant to the task requirements) can be congruent with either the correct or the incorrect response. The latter case causes conflict that results in slower and more error-prone responding (Eriksen and Eriksen, 1974; Lu and Proctor, 1995; MacLeod, 1991), and produces signatures of competing motor plans in the LRP that are similar to those found here (Gratton et al., 1988). Prominent accounts of these tasks posit that an automatic processing of the stimulus happens in parallel with the controlled (decision) process (Servant et al., 2016; Ulrich et al., 2015) It is plausible that the LRP ‘bolus’ in our study could arise from a related mechanism in which the value cue automatically ‘primes’ a response, although it seems likely that value-biased responding is more intentional since it may confer a benefit in terms of the increased reward. Indeed, the patterns of biased anticipatory motor preparation we see in this study can not be present in tasks where the conflict does not arise until after stimulus onset; in such tasks the anticipatory mu/β buildup activity while present is unbiased (Feuerriegel et al., 2021). In the case of these β signals, the fact that the buildup happens earlier under speed pressure (Kelly et al., 2021) suggests that they are much more likely to be strategic rather than automatic, and we would not expect a bottom-up lateralization in response to the physical appearance of the cues due to their symmetric design. Nonetheless, even if different in nature, some of the functional dynamics arising from our value bias cues are interestingly similar to those arising from conflict tasks where both competing influences are externally presented.”

Fourth, given that some of the authors previously made a strong case that evidence accumulation is reflected in the centro-parietal positivity (CPP), it is a bit surprising not to see any mention of this notion in the current manuscript, and instead only read about motor β activity and lateralized readiness potentials. There are likely reasons for the lack of analysis of the CPP, but this would deserve some justification in the revised manuscript.

Many readers are indeed likely to have this thought and we agree it is important to justify. In this study, a compromise that had to be made to ensure fast stimulus-triggered evidence accumulation in this task similar to the majority of decision tasks with sudden-onset. This gives rise to visual evoked potentials that obscure the CPP (Kelly & O’Connell, 2015, *Journal of Physiology-Paris, review article*). We have added an explanation where we describe the decision variable simulations (line 385):

"While our previous work compared the absolute value of the simulated cumulative bias and evidence function (x(t)) to the centroparietal positivity (CPP)—an event related potential thought to be related to evidence accumulation (Kelly et al., 2021)—here this component was obscured by large potentials evoked by the sudden stimulus onset, and thus could not be reliably used in the same way."

3) While the general modeling framework is innovative and highly appealing, there are modeling issues with the analyses reported in the manuscript. A first potential limit of the findings is the sample size and composition. With only 18 participants (one of which was a co-author of this manuscript, see point 4 below), the robustness of the authors' modeling results remains an open question. This low sample size is particularly problematic because the modeling approach used by the authors currently ignores between-participant variability and relies on a fixed-effects approach. It is unclear why this was done (rather than fitting data separately per participant or fitting the data using a hierarchical model), because it induces substantial variance in the fits caused by between-participant differences. This should be corrected in the revised manuscript. Furthermore, fitting the model separately for each participant (possibly using a hierarchical approach) would allow testing if the best-fitting model identified in Table 2 provides the best account of the data for most participants. In practice, we suggest fitting the model separately for each participant and to report for how many participants each of the implemented models provided the best account of the data. Moreover, we would suggest reporting Akaike weights (or relative likelihoods) as described in Wagenmakers & Farrell, 2004.Wagenmakers, E.-J., & Farrell, S. (2004). AIC model selection using Akaike weights. Psychonomic Bulletin & Review, 11(1), 192-196. https://doi.org/10.3758/BF03206482

We have included Akaike weights in the revised manuscript as suggested (Table 2, methods line 651).

We thank the reviewers for this general point about accounting for individual variability in our modeling, which we did not go into sufficient detail on in the paper and address much more comprehensively now in our revised paper. The reason we did not fit the models to individual subjects is that, in contrast to models solely fit to behavior where each individual’s data can be taken as an accurate reflection of the outcomes of their true individual decision process, our neurally-constrained models constrain certain key parameters to equal EEG beta amplitude measures, which are much less reliable on an individual-subject level. We conduct the modeling on a grand-average level because grand-average β-amplitude dynamics are much more robust, whereas at the individual level, it is not unusual to have certain individuals showing no signal at all due to factors independent of the decision process such as brain geometry. We now make these reasons for grand-average level modeling clearer (lines 390, 458). However, even if the modeling approach forces us to do this, it is still critical for us, for the reasons the reviewers highlight, to take every possible measure to ensure that the grand-average behavioral and neural data used for the modeling reliably represent the same qualitative patterns present to varying extents in the individuals, to safeguard against any spurious effects arising due to individual variability. We already address this to some extent – for example to ensure grand-average behavior does not represent a spurious mix of patterns that are each unique only to certain individuals, we average data quantiles as opposed to pooling the data and then taking the quantiles. To go further in demonstrating grand-average data reliability, we have also now added 3 new analyses, which we describe in turn:

First, to increase our confidence that the group-averaged quantiles provide a valid characterization of the behavioral data, we added a new analysis to the paper, following Smith & Corbett (2019) (line 107):

“Our ultimate goal was to develop a model that could jointly explain the group-average EEG decision signals and behavior. Behavior was quantified in the RT distributions for correct and error responses in each stimulus and value condition, summarized in the 0.1, 0.3, 0.5, 0.7 and 0.9 quantiles (Ratcliff and Tuerlinckx, 2002). Following the analysis of Smith & Corbett (2019), we verified that the individual RT quantiles could be safely averaged across participants without causing distortion by plotting the quantiles of the marginal RT distributions for the individual participants against the group-averaged quantiles, for each of the 8 conditions (Figure 1—figure supplement 1). The quantiles of the individual distributions were seen to fall on a set of very straight lines, indicating that the quantile-averaged distribution belongs to the same family as the set of its component distributions (Smith and Corbett, 2019), thus approximating the conditions for safe quantile-averaging identified by Thomas and Ross (1980). We calculated the Pearson correlations between each individual’s quantiles and the group average with that individual excluded, for each condition (see Figure 1—figure supplement 2), finding that the lowest r^2^ was 0.965 while most values were above 0.99. These analyses indicate that quantile-averaging will produce a valid characterization of the pattern of behavioral data in the individuals.”

As noted above, further confidence in our model comparison results is provided by the fact that the main defining aspects of the best-fitting model – gradually increasing evidence and an initial burst of value bias – are individually consistent with findings in the existing literature in which model comparisons were performed at the individual level (Smith & Lilburn, 2020, Diederich & Busemeyer, 2006, Afacan-Seref et al., 2018).

Second, to further verify the validity of the key neural data finding of bias towards the low value alternative in the anticipatory β buildup–which has not been found previously and featured in the neural constraints common to almost all of the models we tested–we now display the individual data values for β buildup rate in Figure 2—figure supplement 1. Note that slopes are negative because β is a decreasing signal. This shows that despite the fact that absolute levels of β amplitude vary quite widely across the group as is typical in human EEG, the majority of individuals (14 out of 17) do show the steeper buildup for the low-value alternative, and it is not the few individuals with particularly strong β amplitude that also dominate in terms of the additive effect of low vs high value buildup rate. Thus, like the behavioral data, the grand-average neural data used for fitting the models are also representative of what is happening in the individuals. In the text we note (line 159):

“These β slopes for the high and low-value alternatives, averaged across conditions, are shown for each individual in Figure 2—figure supplement 1. Despite the fact that absolute levels of β amplitude vary quite widely across the group as is typical in human EEG, the majority of individuals (14 out of 17) show steeper buildup for the low-value alternative.”

Finally, we have also added a Jackknifing procedure to the model comparison to assess its sensitivity to participant selection (line 390):

“Jack-knifing Procedure for Model Comparison. The variability in individual-participant EEG precluded us from performing neurally-constrained modeling at the individual level, so it was not possible to verify that this model comparison would hold for all participants. While the analysis represented in Figure 1—figure supplements 1 and 2 reassured us that the quantile-averaging of the data did not cause distortion, we nevertheless sought to take a step towards quantifying how much our participant selection affected the model comparison results. To this end, we repeated the model comparison for the 4 main neurally-constrained models and the DDM 17 times in turn with one participant excluded each time. The BurstIE model was strongly preferred for all of the samples (see Figure 4—figure supplement 5).”

4) Data curation: it would be safer to exclude the data from the one participant who was also a co-author of this study. Similarly, it would be safer to to not include the data from the second accidentally recorded experimental session in the data to rule out training effects.

As this is a within-subjects design, the overall findings cannot be biased by some participants having more trials – they still get an equal voice in the sample. As the reviewers point out in another comment, the sample size is not extremely large and therefore it was appropriate to not exclude data unless there is a concrete reason to believe their inclusion could bias the results. There was no reason in this case to suspect that a participant who is privy to the purpose of the study could bias the results, especially given that our goal was simply a complete characterisation of the dynamics of value biasing from anticipation through to the response in people seeking to optimise performance, and that the main novel effect of steeper anticipatory buildup for the low-value alternative was certainly not something we would have predicted let alone hoped for. In addition, the jackknifing results described above demonstrated that removing either of these participants does not alter the findings substantively. We have therefore opted to keep all participants and sessions in the main dataset. Note that, despite this, the figures and statistics have all been slightly updated because an EEG data file for one block for one participant which was originally missing has since been retrieved and included.

5) The choice of the experimental design could be better motivated in the introduction. You write in the introduction that you "primarily focused on the value biasing dynamics in common across these challenging regimes". So it feels like task conditions with distinct value differences could have been more instructive than the proposed experimental design. A design with distinct value differences would also distinguish between your favored account (where different levels of value should parametrically affect motor β) and a simple priming account (where value should not matter). The revised manuscript should better explain why the chosen experimental conditions are important for revealing the main findings.

The purpose of running this particular set of regimes was to be able to check whether the dynamics of value biasing – some elements of which we had observed in previous work only employing a single-difficulty, highly discriminable color discrimination task under high speed pressure – generalised or differed across task contexts that were equally challenging but for different reasons (i.e. longer deadline but harder discrimination) and in regimes with multiple difficulties (i.e. our interleaved regime). The latter factor of heterogeneous versus homogeneous stimulus discriminability is important because drift rate biases have been shown to be optimal–in the context of the diffusion decision model–for the former case but not the latter (Moran 2015, Hanks 2011). As it turned out, the behavioral and neural data patterns were similar across these regimes and our modeling efforts were thus focused on capturing the interesting multiphasic pattern in common across all of them. We now clarify this motivation in the revised paper (line 93), and add an explanation for our use of a single value differential:

“These regimes were similarly challenging but in different ways, allowing us to further explore the extent to which the uncovered value biasing dynamics generalize across task contexts where the demands are placed through lower discriminability versus through tight deadlines, and where stimulus discriminability is heterogeneous versus homogeneous (Hanks et al., 2011; Moran, 2015). […] We imposed a single value differential (30 vs 10 points) that, combined with the deadline and coherence settings, induced a decision-making approach that was guided strongly by both sensory and value information.”

We agree that parametrically manipulating value differential would be of interest in future work, especially given that interesting qualitative shifts in strategy might be observed: for example in a past study of value-biased decisions (Blangero & Kelly 2017) in which a much greater value differential was employed for the purposes of identifying a relative-value encoding ERP signal, we observed that some subjects ignored the sensory information entirely and consistently made action choices toward the high-value alternative regardless of sensory evidence. Interestingly a task version with an extremely tight deadline had a similar impact on the overall group, where participants again ignored sensory evidence and quickly executed a pre-programmed, value-based response. While these cases illustrate the large and interesting differences that can be uncovered using extensive manipulations of factors contributing to task difficulty, our current focus was instead on characterizing in greater detail the dynamics of value biasing in a set of situations where the subject must truly draw on both value and sensory information in concert.

6) A Bayes Factor (BF) of 0.87 does not afford the conclusion that there was no significant difference between low-coherence conditions. Instead, it indicates that the data do not contain enough evidence to draw any conclusions. The manuscript should be corrected accordingly.

We have updated the manuscript (caption of Figure 1):

“The low-coherence interleaved condition was slightly more accurate than the low coherence blocked condition but not significantly so, though the Bayes factor indicates the data contain insufficient evidence to draw definite conclusions (p=0.1, BF10=0.87).”

[Editors' note: further revisions were suggested prior to acceptance, as described below.]

You have provided detailed responses to the essential revisions outlined in our former decision letter. We had now time to consider your responses, we are sorry for the delay and thank you for your patience.We found that you provided convincing and in-depth answers and changes to the manuscript. In particular, you now test your model against the classical drift-diffusion model (DDM) found in the literature. The fact that the dynamics of β amplitude vary as a function of speed pressure makes it unlikely that it reflects automatic priming. The paper also reads better now that you have added several additional explanatory paragraphs where needed, and you provide a more accurate description of the existing literature.Two responses in your point-by-point rebuttal letter required more thought: 1. your decision to stick to model fits in terms of group-level averages rather than individual participants, and 2. your decision not to exclude the participant who was not naive to the purpose of the study.Regarding point 1, we found that you indeed provided convincing explanations as to why you did not follow the reviewers' suggestion to perform individual fits to the data. Nevertheless, as a minor revision request, I would find it more appropriate that you introduce the fact that you will be fitting group-level averages rather than individual participants at the beginning of the "Model fits" section (starting at line 316). You should describe in a few sentences the rationale (which you outline clearly in the rebuttal letter) as to why you are using such a fitting approach over the more standard (individual participant) approach. You make totally valid points in the rebuttal letter and we think they should be spelled out at the onset of the "Model fits" section. These are important points for readers. It is good that you now provide the "jackknifing procedure for model comparison" subsection at the end of the Results section which, we agree with you, provides additional and supporting evidence that your results are robust and not driven by an individual subject.Regarding point 2, we tend to disagree that your study of biasing effects does not create issues when testing participants who are not blind to the purpose of the study. Although the biases you describe may not be explicit and fully voluntary, being informed about the purpose of the study may (even if unlikely) create effects that are not present in naive subjects. Because reviewers specifically asked for you to remove the participant in question, we took some time to consider your decision to keep this participant – in light also of your response to point 1 above. With the additional data that you report in the revised manuscript, and in the rebuttal letter, we agree that it is highly unlikely that a single participant drives the group-level effects that are reported in the study. Therefore, we agree that this non-naive participant can be kept in the dataset.

As requested, we have included the following explanation for the group average fitting approach at the beginning of the ‘Model Fits’ section:

“We did not fit the models to individual subjects because, in contrast to models solely fit to behavior where each individual’s data can be taken as an accurate reflection of the outcomes of their true individual decision process, our neurally-constrained models constrain certain key parameters to equal EEG β-amplitude measures. These EEG measures are much less reliable on an individual-subject level, where it is not unusual to have certain individuals showing no signal at all due to factors independent of the decision process such as brain geometry. We therefore conduct the modeling on a grand-average level because grand-average β-amplitude dynamics are much more robust.”